# Cooperative clamp-mediated promoter recognition by poxviral RNA polymerase and its TBP/TFIIB-like partner

Stefan Jungwirth [1,4], Julia Bartuli[1], Stephanie Lamer[2], Andreas Schlosser[2], Clemens Grimm [1,4] ✉ & Utz Fischer [1,3] ✉

The recruitment of RNA polymerase to gene promoters is a critical step in gene expression. For RNA polymerase II, this process is initiated by TBP and TFIIB, with homologs of these TBP/TFIIB pairs found in all known multi-subunit RNA polymerase systems. Here, we describe a mode of promoter recognition by the poxviral intermediate transcription factor 3, VITF-3. This heterodimeric factor comprises an atypical TBP/TFIIB pair forming a stable ring structure inert towards DNA in the absence of viral RNA polymerase. Promoter recognition instead requires concerted VITF-3 and viral RNA polymerase binding, as shown by cryo-EM analysis of the intermediate pre-initiation complex. During the formation of this complex, viral RNA polymerase facilitates ring opening and loading of VITF-3 onto the promoter, anchoring the polymerase at the transcription start site. Our findings suggest viral RNA polymerase could act as a clamp loader for VITF-3 and identify VITF-3 as an unusual TBP/TFIIB pair.

The initiation of transcription is a fundamental process across all domains of life, requiring the assembly of a pre-initiation complex (PIC) at precisely defined regions within gene promoters[1]. This complex contains the DNA-dependent RNA polymerase (RNAP) and transcription factors (TFs) positioned on the promoter, but it remains transcriptionally inactive at this stage. Eukaryotic and archaeal multi-subunit RNAPs, including RNAPs I, II and III utilize a core set of two universally conserved TFs for transcription initiation[2–5]. One of these is TATA-box binding protein (TBP), which binds the TATA box of RNAP II/III genes and connects the polymerase to its cognate promoter[5]. TATA-box binding is mediated by a conserved domain comprising two repeats that fold into a saddle-like structure. This domain interacts with DNA via the minor groove, inducing bending of the duplex to recruit additional factors and stabilize the PIC[6,7]. For transcription initiation at RNAP I promoters, TBP is not strictly required for basal activity, as it functions primarily as a coactivator that stabilizes promoter interactions[8]. To date, cryo-EM structures have only been determined for the basal RNAP I PIC, in which TBP is absent[9].

Nevertheless, TBP is essential to achieve wild-type transcription rates of RNAP I[10]. In addition to genuine TBP, several TBP-related proteins or TBP-like factors (TBPLFs) have been identified in metazoans, but their mode of action in transcription initiation is understood in only a few cases[11–13].

The second universally conserved TF belongs to the TFIIB-related protein (TFBRP) family. Its name-giving member TFIIB serves multiple functions in RNAP II transcription, including stabilizing the DNA–TBP complex and assisting RNAP II recruitment to the promoter[14–16]. Additionally, TFIIB contributes to promoter recognition, DNA melting, and stabilization of the template strand at the active site via the B-reader[17]. Other TFBRPs perform similar tasks but employ interactions that differ partially or entirely from the canonical RNAP II system[18,19].

Recent studies on poxviruses provided unexpected insight into the evolution of TBP/TFIIB pairs involved in the transcription by multi-subunit RNAPs[20–22]. Poxviruses are double-stranded DNA viruses that replicate and express their genomes in the cytoplasm of their hosts

[1]Department of Biochemistry 1, Theodor Boveri-Institute, University of Würzburg, Würzburg, Germany. [2]Rudolf Virchow Center for Experimental Biomedicine, University of Würzburg, Würzburg, Germany. [3]Helmholtz Institute for RNA-based Infection Research (HIRI), Helmholtz Center for Infection Research (HZI), Würzburg, Germany. [4]These authors contributed equally: Stefan Jungwirth, Clemens Grimm. ✉e-mail: clemens.grimm@uni-wuerzburg.de; utz.fischer@uni-wuerzburg.de

using their own transcription and replication machinery[23–25]. Structural and functional investigations on the prototypic poxvirus Vaccinia have revealed an elaborate transcription apparatus centered around a virus-encoded multi-subunit RNA polymerase (vRNAP)[21,26,27]. This complex, composed of eight subunits (Rpo147, Rpo132, Rpo35, Rpo30, Rpo22, Rpo19, Rpo18, and Rpo7), transcribes all poxviral genes[26,28–30]. For co-transcriptional RNA processing and transcription termination, vRNAP cooperates with further virus-encoded factors including the hetero-dimeric capping enzyme (CE)[22,31–35].

During the viral replication cycle, vRNAP is directed to specific promoters through specialized transcription and processing factors, enabling the coordinated expression of early, intermediate, and late genes[26,36]. Transcription initiation at early promoters requires the heterodimeric Vaccinia early transcription factor (VETF-l and -s) and Rap94 to assemble the early pre-initiation complex (ePIC)[20,37–42]. Rap94 is a multifunctional protein containing a TFIIB homology domain, whereas VETF-l possesses a unique TBP-like domain (TBPLD), defining VETF-l and Rap94 as a TBP/TFIIB pair[20,21]. Unlike the symmetric binding of TBP to the minor groove of the TATA-box, the TBPLD of VETF-l recognizes early gene promoters by asymmetrically contacting the major groove[7,20,21]. The CE is not part of the ePIC but assembles with the elongating vRNAP for co-transcriptional capping and termination[22,43].

Transcription of intermediate genes relies on a distinct set of virus-encoded factors that act together with vRNAP[26]. Intermediate promoters comprise the consensus-lacking, but AT-rich core element separated by a 12-base pair (bp) spacer from the initiator element, which contains the strictly conserved TAAA motif spanning positions −1 to +3[44,45]. Several intermediate TFs have been identified, including CE, Vaccinia intermediate transcription factor 3 (VITF-3) consisting of subunits VITF-3s and VITF-3l as well as the host factors G3BP1 and p137[46–49]. Biochemical studies suggested that the VITF-3s/l heterodimer recognizes the AT-rich core element, but its precise role in inter-mediate PIC (iPIC) assembly and its mode of function in transcription remains unknown[45,48].

In this study, we establish a strategy to isolate Vaccinia inter-mediate transcription complexes enabling their investigation by bio-chemical and structural approaches. We identify the heterodimeric VITF-3 as a TBP/TFIIB pair that acts in a radically different manner than other known TBP/TFIIB pairs. The cryo-EM structure of the iPIC shows that the VITF-3 heterodimer forms a closed ring around the AT-rich upstream element, anchoring the vRNAP and the viral CE at the pro-moter. The structural dissection of PIC assembly revealed an unchar-acterized mechanism that requires VITF-3 loading onto the intermediate promoter and direct promoter recognition facilitated by vRNAP.

## Results

### Purification of the Vaccinia intermediate pre-initiation complex

To investigate poxviral intermediate gene expression we modified a previously established method for the isolation of vRNAP complexes from infected HeLa S3 cells[21,50]. This procedure relies on the recom-binant Vaccinia strain GLV-1h439 expressing the FLAG-tagged vRNAP subunit Rpo132 and allows for the purification of early transcription complexes (i.e., core and complete vRNAPs, Fig. 1A, lane 1)[21]. To selectively enrich intermediate transcription complexes, we modified our purification protocol by infecting cells in presence of the replica-tion inhibitor cytosine β-D-arabinofuranoside (AraC) prior to vRNAP purification[51]. This treatment prevents the progression of viral infec-tion into the replicative stage but allows early transcription and thus enrichment of early gene products including both subunits of VITF-3. vRNAP was affinity-purified from AraC-treated infected cells and its composition was compared with vRNAP isolated from untreated infected cells by SDS-PAGE and mass spectrometry (Fig. 1A, lanes 1 and 2; Fig. 1B; see also Supplementary Fig. 1A for the investigation of the purified complex by gradient centrifugation). vRNAP core subunits

and the heterodimeric CE were equally enriched in both purifications, showing that AraC treatment did not interfere with the expression of early transcripts including those encoding the subunits of the core polymerase. However, the early TFs VETF-s/l, Rap94, NPH-I and E11, all of which are encoded by late genes, were selectively depleted from vRNAP-complexes purified from AraC-treated cells. Instead, vRNAP complexes contained both VITF-3 subunits albeit in sub-stoichiometric amounts (see Fig. 1A for the analysis of isolated vRNAP complexes by SDS-PAGE and Fig. 1B for their analysis by mass spectrometry). The purified complex consisting of vRNAP and CE will be referred to as vRNAP^AraC throughout this manuscript. Since vRNAP^AraC and VITF-3 did not form a stable complex on their own, we sought to assemble both units on an intermediate promoter scaffold. To this end, we first gen-erated a sequence motif consensus of the intermediate promoter, which revealed an AT-rich core element and a characteristic TAAA motif at the TSS, consistent with previous descriptions (Fig. 1C)[44,52,53]. Based on this analysis, a 70 bp intermediate promoter scaffold was added to lysate of AraC-treated HeLa cells prior to complex purifica-tion (Fig. 1D). This allowed for the efficient isolation of a stable and stoichiometric complex composed of vRNAP, VITF-3, CE and promoter scaffold (Fig. 1A, lane 3, see also Supplementary Fig. 1B for the analysis of this complex by gradient centrifugation). Based on its composition, the purified complex constitutes a bona fide iPIC or an intermediate initial transcribing complex (iITC), capturing the transcription machinery before or immediately after synthesis of the first RNA nucleotides.

### Structure of the iPIC at 2.4 Å resolution

We analyzed the purified vRNAP complex by means of cryo-EM. A dataset containing about 9 million particles allowed for a 2.4 Å reconstruction and building of an almost complete model (Fig. 2A, Table 1 for data collection, reconstruction and model refinement sta-tistics; see also Supplementary Fig. 2). The reconstruction shows density for core vRNAP at very high (<2 Å local) resolution including the promoter DNA from position −30 to +14, as well as the VITF-3 heterodimer and CE. The DNA is bound deep inside the vRNAP cleft with the transcription start site (TSS) in close vicinity to the active site. Since a DNA/RNA hybrid is clearly absent, the structure represents the iPIC rather than an iITC. The bubble region is centered around the active site with both nucleic acid strands widely separated indicating an open complex conformation (Fig. 2B). Remarkably, the hetero-dimeric VITF-3 forms a ring enclosing the upstream AT-rich element at the promoter from −25 to −15 and guides the DNA into the vRNAP cleft by introducing a 90° kink in the helix axis. VITF-3 is supported by the vRNAP wall and protrusion domains on one side and by the CE on the other, leading to a rugged integration into the iPIC.

While being tightly associated with the iPIC, the CE subunits D1 and D12 are divided into two modules (Fig. 2A, C). The first consists of the N-terminal part of D1, whereas the second module holds the C-terminus of D1 bound to D12. We observed in the 3D classes of the iPIC that module 1 always occupies the same binding surface with Rpo147 and Rpo18, whereas the density of module 2 is either con-tacting VITF-3s or remains flexible (Fig. 2C, D; see also Supplementary Fig. 2A). The letter resulted in a particle subset lacking module 2, which we termed iPIC_CEm.

### The iPIC structure resembles the co-transcriptional capping complex

Structural comparison of the iPIC with previously resolved vRNAP assemblies showed that it most closely resembles the co-transcriptional capping complex (CCC) that forms during early tran-scription (compare Fig. 3A and B)[20,22]. Both complexes consist of all vRNAP subunits, DNA, and the CE heterodimer, but only the CCC contains nascent mRNA. In the latter, module 2 of the CE interacts with the upstream DNA and Rpo132 on one side of the cleft, while module 1

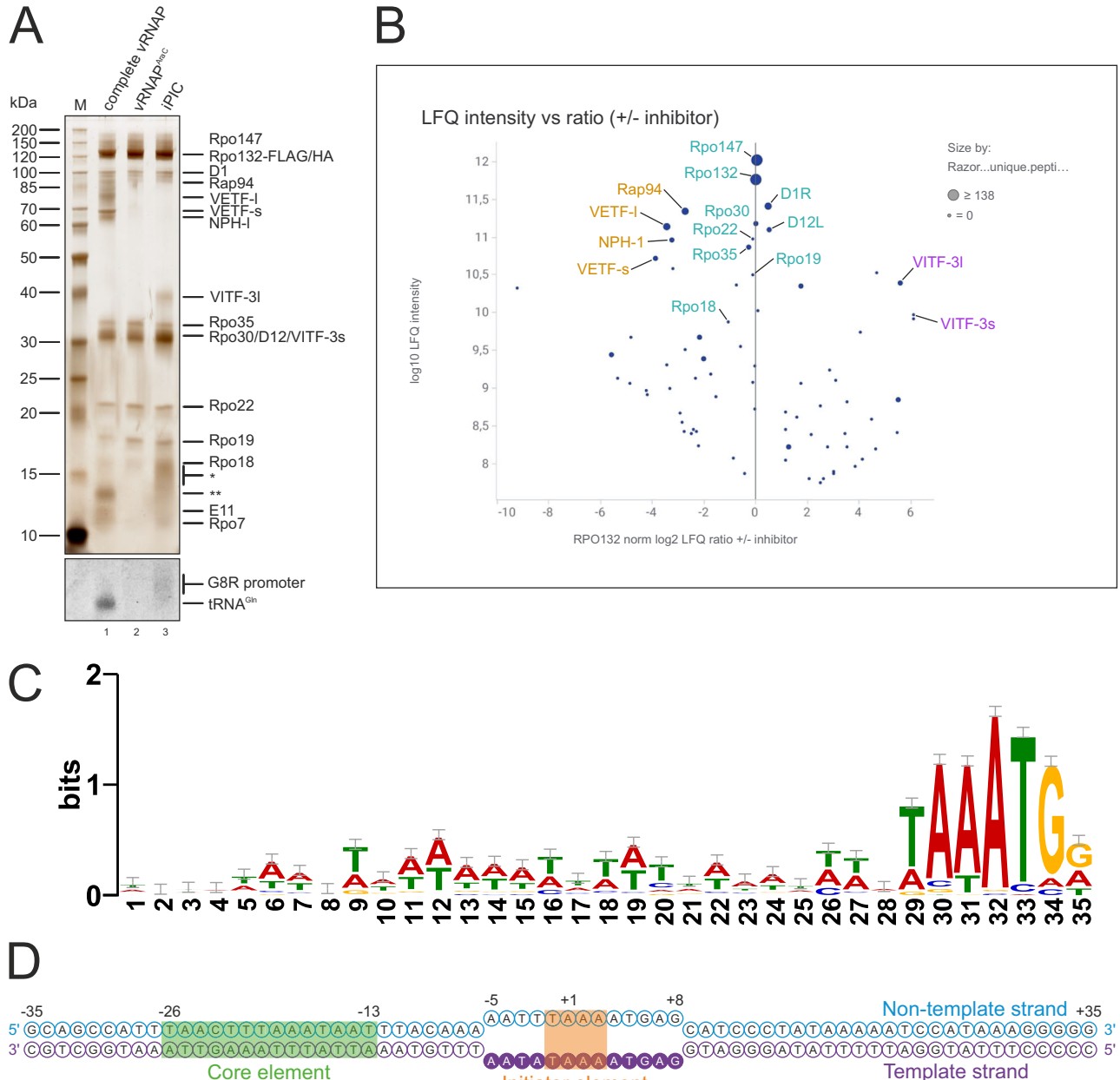

**Fig. 1 | Purification of the iPIC. A** Protein composition of complete vRNAP (lane 1), vRNAP^AraC (lane 2) and the iPIC (lane 3). Nucleic acid staining is shown below; a molecular weight marker is shown on the left and respective proteins are indicated on the right. In the protein staining, the 70 bp *G8R* promoter scaffold is indicated with one asterisk, whereas two asterisks show tRNA^Gln. Source data are provided as a Source data file. **B** Mass-spectrometry based protein comparison of FLAG-IP eluates from GLV-1h439 infected HeLa S3 cells in presence and absence of AraC during infection. The log₁₀ LFQ intensity is plotted against the log₂ ratio of +/− AraC normalized on the FLAG-tagged subunit Rpo132. The early transcription factors Rap94, VETF-s/l, and NPH-I are highlighted orange, the intermediate transcription

factor subunits VITF-3s/l are depicted purple and the vRNAP core subunits as well as the capping enzyme D1/D12 shown in cyan. **C** Motif logo of the Vaccinia intermediate promoter. The motif was generated by the program MEME using 48 annotated Vaccinia intermediate promoter sequences spanning 50 nt upstream and 4 nt downstream of the start codons. The search parameters were one motif per sequence and a motif length of 35 to 38 nt. This figure was adopted from Yang et al. (2011) and Deng et al. (2025)[52,53]. **D** The 70 bp DNA scaffold based on the *G8R* promoter ranging from −35 to +35 in respect to the TSS used for the reconstitution of the iPIC. The core element from −26 to −13, initiator element from −1 to +3, as well as the mismatch bubble on the template strand from −5 to +8, are indicated.

stably associates with Rpo147 on the other side. In the iPIC, module 1 occupies a similar position as in the CCC but module 2 is displaced upward, as its binding interfaces on the upstream DNA and Rpo132 are occupied by VITF-3. As a result, module 2 either contacts the VITF-3 dimer or remains flexible (Fig. 3A, B; see also Fig. 2). Interestingly, the RNA exit channel that accommodates the nascent RNA chain in the CCC is occupied in the iPIC by the N-terminus of VITF-3s, which constitutes a TFIIB reader-like element (BRLE, see also below). The overall structural similarity between the iPIC and CCC suggests that the iPIC

transitions into the CCC upon initiation of transcription and promoter escape, which is further supported by biochemical experiments shown below. In contrast, the iPIC differs significantly from the ePIC in structure and composition (Fig. 3B, C)[20]. In the ePIC, promoter recognition is mediated by the VETF-s/l heterodimer bridging the polymerase cleft, as well as by Rap94, which is wrapped around vRNAP. Unlike the iPIC, the CE is not included in the ePIC but re-associates during initial transcription to facilitate the transition into the CCC[20]. Early and intermediate TFs hence adopt distinct conformations and

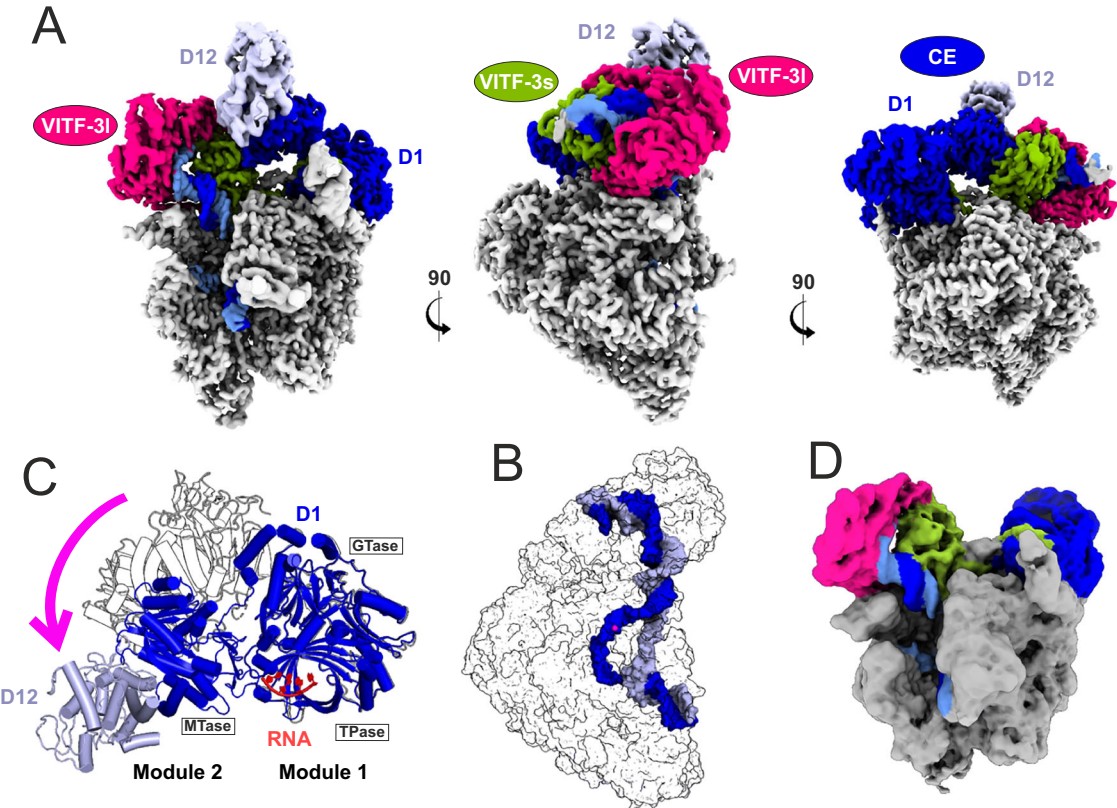

**Fig. 2 | Structure of the iPIC at 2.4 Å resolution. A** Three orthogonal views of the iPIC cryo EM density enhanced with DeepEMhancer[78]. The core viral RNA polymerase is colored gray, VITF-3s/l are green and magenta, respectively, the CE subunits D1/D12 are shown in dark- and lightblue and template and non-template strands are depicted darkblue and blue. This color code remains consistent throughout the figures. **B** The protein content of the iPIC shown as transparent accessable surface model and the solvent-accessible surface of the DNA. Disordered DNA regions were reconstructed manually. The active site $Mg^{2+}$ is marked by a magenta sphere. The orientation resembles the middle view of the iPIC in (**A**). **C** Transition of the CE structure found in the iPIC (black outline) into the CCC structure (colored cartoon) by a movement of module 2. The three catalytic centers and the positon of synthesized RNA are indicated. **D** Cryo-EM density of the iPIC$_{CEm}$-subset lacking density for module 2 of the CE.

engage in different interactions, contributing to the pronounced structural differences between the two Vaccinia PIC structures.

## vRNAP, CE and VITF-3 enable intermediate transcription and capping

We next investigated whether the assembly of the iPIC leads to productive transcription in an intermediate promoter-dependent manner. To assess this question, we performed in vitro transcription assays on templates under the control of early, intermediate, and late promoters using isolated complete vRNAP or vRNAP$^{AraC}$ in presence or absence of recombinant VITF-3 (Fig. 4A, B). As expected, complete vRNAP, which constitutes an "all in one" unit combining vRNAP with early transcription and processing factors, efficiently transcribed the template under control of the early promoter (Fig. 4B, lane 1), whereas no transcription was observed on templates with intermediate (*G8R*) or late (*A9L*) promoters (Fig. 4B, lanes 4 and 7). In contrast, vRNAP$^{AraC}$ failed to transcribe templates with early and late promoters (Fig. 4B, lanes 2 and 8). However, it enabled intermediate transcription albeit at a very low level (Fig. 4B, lane 5), likely due to small amounts of VITF-3 in the vRNAP$^{AraC}$ preparation (see Fig. 1B). Importantly, the addition of recombinant VITF-3 selectively boosted the transcription from intermediate promoters by at least 10-fold (Fig. 4B, lane 6), indicating that vRNAP, CE and VITF-3 are necessary and sufficient for robust transcription from intermediate promoters in vitro.

Vaccinia transcripts acquire an m7GpppN cap structure when produced in infected cells and co-transcriptional capping has been shown to occur in vitro using complete vRNAP[22,49]. To investigate whether co-transcriptional capping also occurs during intermediate transcription, [32P]-labeled mRNAs were generated by complete vRNAP, vRNAP$^{AraC}$ supplemented with recombinant VITF-3 and bacteriophage T7 RNAP as a control. The [32P]-labeled transcripts were pooled and analyzed by immunoprecipitation using monoclonal H-20 antibodies that specifically recognize m7G-cap structures (Fig. 4C)[54]. Early and intermediate transcripts were immunoprecipitated to approximately 25% and 90%, respectively, whereas the uncapped T7 transcript remained quantitatively in the supernatant (Fig. 4C, D). These results demonstrate that isolated early and intermediate viral transcription complexes efficiently generate m7G-capped transcripts in vitro thus faithfully recapitulating the in vivo situation.

## vRNAP assists in loading the ring-shaped VITF-3 onto the promoter

To investigate the mechanism of intermediate promoter recognition, we closely examined VITF-3 and its binding to DNA. Sequence analysis and structural studies showed that VITF-3 constitutes an unusual TBP/TFIIB pair with VITF-3l being the TBP-related subunit and VITF-3s the TFIIB-like subunit (Fig. 5A). The TBP-related saddle-like fold in VITF-3l enables the binding of upstream promoter DNA. However, unlike canonical TBP, it features extended "saddle flaps" that allow simultaneous interaction with both sides of VITF-3s to form a ring-shaped complex that encircles the promoter (Fig. 5B). VITF-3s contains a flexible N-terminal tail of 50 amino acids, which harbors a BRLE. As revealed by the iPIC structure, VITF-3 not only embraces the upstream core element but also induces a bend in the DNA helix axis of

**Table 1 | Cryo-EM data collection, single-particle reconstruction and model refinement statistics**

| | iPIC$_d$ (EMDB-17334) (PDB 8POJ) | iPIC$_s$ (EMDB-17336) (PDB 8PON) | iPIC$_{CEm}$ (EMDB-17335) (PDB 8POK) |
|---|---|---|---|
| **Data collection and processing** | | | |
| Magnification | 75,000 | 75,000 | 75,000 |
| Voltage (kV) | 300 | 300 | 300 |
| Electron exposure (e⁻/Å²) | 70 | 70 | 70 |
| Defocus range (μm) | −1.2 to −2.2 | −1.2 to −2.2 | −1.2 to −2.2 |
| Pixel size (Å) | 1.0635 | 1.0635 | 1.0635 |
| Symmetry imposed | C1 | C1 | C1 |
| Initial particle images (no.) | 9,318,600 | 9,318,600 | 9,318,600 |
| Final particle images (no.) | 1,421,158 | 1,421,158 | 1,421,158 |
| Map resolution (Å) | 2.39 | 2.58 | 2.64 |
| FSC threshold | 0.143 | 0.143 | 0.143 |
| **Refinement** | | | |
| Initial model used (PDB code) | 6RIE | 6RIE | 6RIE |
| Model resolution (Å) | 2.45 | 2.7 | 2.75 |
| FSC threshold | 0.143 | 0.143 | 0.143 |
| Model resolution range (Å) | 50–2.39 | 50–2.58 | 50–2.64 |
| Map sharpening $B$ factor (Å²) | −90 | −90 | −90 |
| **Model composition** | | | |
| Non-hydrogen atoms | 86474 | 85971 | 73838 |
| Protein residues | 5139 | 5139 | 4536 |
| Ligands | 1 Mg2+; 4 Zn2+ | 1 Mg2+; 4 Zn2+ | 1 Mg2+; 4 Zn2+ |
| **$B$ factors (Å²)** | | | |
| Protein | 56.9 | 112 | 136 |
| Ligand | 97 | 166 | 203 |
| **R.m.s. deviations** | | | |
| Bond lengths (Å) | 0.002 | 0.004 | 0.005 |
| Bond angles (°) | 0.42 | 0.61 | 0.53 |
| **Validation** | | | |
| MolProbity score | 1.43 | 1.64 | 1.58 |
| Clashscore | 4.2 | 7.2 | 8.5 |
| Poor rotamers (%) | 0.66 | 0.97 | 0.96 |
| **Ramachandran plot** | | | |
| Favored (%) | 96.5 | 95.4 | 95.2 |
| Allowed (%) | 3.5 | 3.6 | 4.8 |
| Disallowed (%) | 0.0 | 0.0 | 0.0 |

approximately 90°. Consistent with its DNA-encircling architecture, electrostatic potential mapping showed a uniformly positive charge on the inner surface of the VITF-3 ring (Fig. 5C). DNA binding is mediated by the insertion of aliphatic residues of VITF-3 into the minor groove, without significant groove widening. Specifically, two isoleucine residues intercalate into the DNA at promoter positions −21/−22 (Fig. 5D) and −18/−19 (Fig. 5E), disrupting Watson–Crick base pairing. These interactions are primarily directed at the DNA sugar-phosphate backbone and lack obvious base-specific contacts. Because VITF-3 binding disrupts multiple base pairs, it preferentially associates with AT-rich

regions of the upstream core element, where base stacking is less stable than in GC-rich sequences. This low sequence specificity is consistent with previous studies that identified the upstream core element as an AT-rich stretch lacking a defined consensus motif (Fig. 1C)[44].

The presence of the VITF-3 ring encircling the core element in the iPIC structure raised the question of its assembly in this position. To investigate the loading mechanism, we performed electrophoretic mobility shift assays (EMSAs) using vRNAP$^{AraC}$, recombinant VITF-3, and radio-labeled *G8R* promoter DNA containing a mismatch bubble from −5 to +8 relative to the TSS (Fig. 5F; see also Fig. 1C for the DNA architecture). VITF-3 alone did not bind the DNA, despite the presence of an upstream core element within the scaffold (Fig. 5F, lanes 1–4). In contrast, vRNAP$^{AraC}$ exhibited partial DNA binding (Fig. 5F, lanes 5–7), likely reflecting the general affinity of RNAPs for DNA and in particular for DNA containing a mismatch bubble. In contrast, the presence of both VITF-3 and vRNAP$^{AraC}$ led to iPIC formation, as indicated by a slower-migrating complex (Fig. 5F, lanes 8–10), which was also observed using a closed promoter (Supplementary Fig. 3A). However, when a randomized DNA scaffold retaining the same mismatch bubble was used, this higher-order complex did not form (Fig. 5F, lanes 18–20). Mutating either the core- or the initiator element of the G8R promoter, both led to a drastic decrease in iPIC formation (Supplementary Fig. 3B, lanes 1–15). Consistently, introducing either the wild-type (wt) core- or initiator element into the randomized scaffold were not sufficient to restore wt iPIC formation (Supplementary Fig. 3B, lanes 16–30). This indicates that vRNAP$^{AraC}$ is both necessary and sufficient to load the ring-shaped VITF-3 onto the upstream promoter region in vitro. It further shows that intermediate promoter recognition and iPIC formation requires DNA binding of both VITF-3 and vRNAP$^{AraC}$.

Further evidence for assisted loading mechanism of VITF-3 onto the promoter was obtained from structural studies of the iPIC. The dataset yielded in total three different iPIC conformations including iPIC$_{CEm}$ with mobile CE and iPIC$_d$, which showed deep binding of the DNA within the vRNAP cleft. The third particle subset displayed a slightly shallower binding conformation of the downstream DNA (termed iPIC$_s$). Structural comparison between iPIC$_d$ and iPIC$_s$, which likely represent transitional intermediates in the DNA loading process, suggests a conformational continuum (Supplementary Movie 1). The trajectory supports a concerted loading mechanism, wherein vRNAP$^{AraC}$ cooperatively opens the VITF-3 ring to load it onto the upstream promoter. This process forms a stable iPIC, with the VITF-3 ring clamped around the upstream promoter.

### vRNAP detects the TSS and stabilizes melted DNA strands in the iPIC

Inspired by the findings described in the previous section, we next investigated the interactions underlying sequence-specific promoter recognition. Guided by VITF-3, the upstream DNA is directly inserted into the vRNAP cleft, where the strands are separated and positioned for transcription initiation. Since the promoter DNA around both fork points displayed a well-defined cryo-EM density in the iPIC structure, we were able to analyze the DNA pathway in detail (Fig. 6A for a schematic overview, Fig. 6B). The upstream fork point at position −7 is stabilized by fork loop 1 of the Rpo147 clamp domain and by the VITF-3s fork loop (Fig. 6C), which both are simultaneously inserted into the DNA leading to strand separation. Near the downstream fork point, the melted non-template strand stably binds the lobe domain of core vRNAP (Fig. 6D). We found the previously described initiatior element, a TAAA motif at position −1 to +3 on the non-template strand, being directly read out by the vRNAP core subunits (Fig. 1C). The base specificity is generated by an area centered around the Rpo132 lobe, which contacts the bases of the initiator element via His207. This robust binding of the non-template strand stabilizes the downstream fork

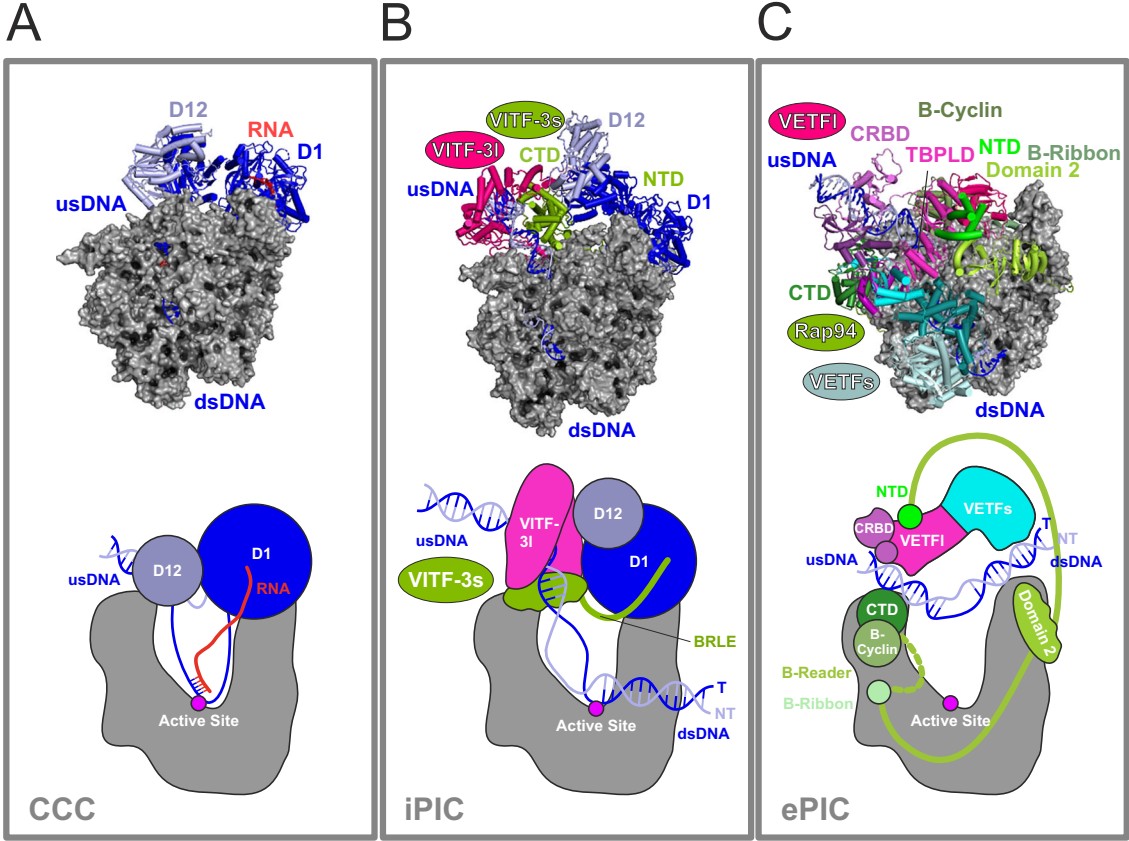

**Fig. 3 | Structural comparison of vRNAP complexes.** The structures of iPIC$_d$ (**B**, 8POJ), CCC (**A**, 6RIE)[22] and ePIC (**C**, 7AMV)[20] were compared. Solvent-accessible surface models of vRNAP complexes with cartoon representations of DNA, RNA and TFs are shown on top. Schemes of corresponding structures are depicted at the bottom. Protein names and functional elements are indicated. usDNA upstream DNA, dsDNA downstream DNA, CTD C-terminal domain, NTD N-terminal domain, TBPLD TBP-like domain, CRBD critical region binding domain, T template strand, NT non-template strand.

point and positions the +1 base of the melted template strand next to the active site. Overall, these resuts indicate that sequence specificity in intermediate transcription is primarily provided by core vRNAP rather than its TFs.

At the downstream fork point at position +4, three residues, Asn450, Val451, and Ile453, from the Rpo147 fork loop 2 play a key role: They separate the DNA bases of the template and non-template strands and protect them from the surrounding solvent. Aside, Asn450 directly contacts the sugar phosphate backbone between position +4 and +3 of the non-template strand, where it makes a sharp turn towards the lobe-bound initiator element. In the opposite direction, the template strand is stabilized by the bridge helix of Rpo147, which is inserted between the bases at positions +3 and +4 (Fig. 6E). Especially Ser750 and Thr754 stabilize the single-stranded template strand. Starting at the upstream fork point where the fork loop of VITF-3s assists in strand separation, the VITF-3s N-terminal tail is embedded along the outside of vRNAP to the catalytically active module 1 of the CE (Fig. 5F). On this pathway, the VITF-3s helix $3_{10}$ and the BRLE are stabilized by interactions with vRNAP and CE. Interestingly, the comparison of iPIC and CCC showed that the N-terminal domain of VITF-3s in the iPIC occupies the same path as the emerging RNA chain in the CCC (Fig. 3A, B), suggesting a potential promoter escape mechanism (see "Discussion")[22].

### Vaccinia transcription factors evolved from canonical TBP

The unique architecture of VITF-3 prompted us to examine its conservation across poxviruses. Multiple sequence alignments of the two Vaccinia VITF-3 subunits showed only minor differences among other

*Orthopoxviruses* and moderate divergence in comparison to the crocodile poxvirus (Supplementary Fig. 4A, B). Both VITF-3 subunits are conserved throughout the *Chordopoxvirinae*, the *Poxviridae* subfamily infecting vertebrates, whereas no homologs were detected in the insect-infecting *Entomopoxvirinae* (Supplementary Fig. 4C). Because the Nile crocodile poxvirus represents one of the most distantly related members of the *Chordopoxvirinae*, we assessed whether its VITF-3 complex shares structural and functional similarity with its counterpart in Vaccinia. AlphaFold predictions for both complexes revealed highly similar overall architectures (Supplementary Fig. 4D, E). Together, these analyses show that both VITF-3 subunits are conserved within the *Chordopoxvirinae* and likely perform equivalent functions across all members of this subfamily.

The structures of iPIC and ePIC shed light on the evolutionary trajectory of the TBP fold in the context of cellular and viral PICs. The ancestral TBP prototype likely resembled a monopartite RNA-binding fold as seen in RNase HIII, which is composed of five antiparallel β-strands and two α-helices (Fig. 7A, B)[55]. Canonical TBP evolved through gene duplication into a bilobal, pseudo-symmetric structure, with each lobe containing five β-strands and two α-helices[7]. As general TF, TBP plays a central role in the transcription systems of RNAP II, RNAP III, and archaea, where it binds promoter DNA, induces bending, and recruits TFIIB or its homologs[5,7]. Strikingly, TBP does not directly interact with the RNAP core in these systems but rather recruits TFIIB or its homologs for this purpose (Fig. 7C)[5].

In the context of Vaccinia early transcription, the domain architecture of the bipartite TBP homolog VETF-l differs from that of canonical TBP by featuring an extended hinge region that connects

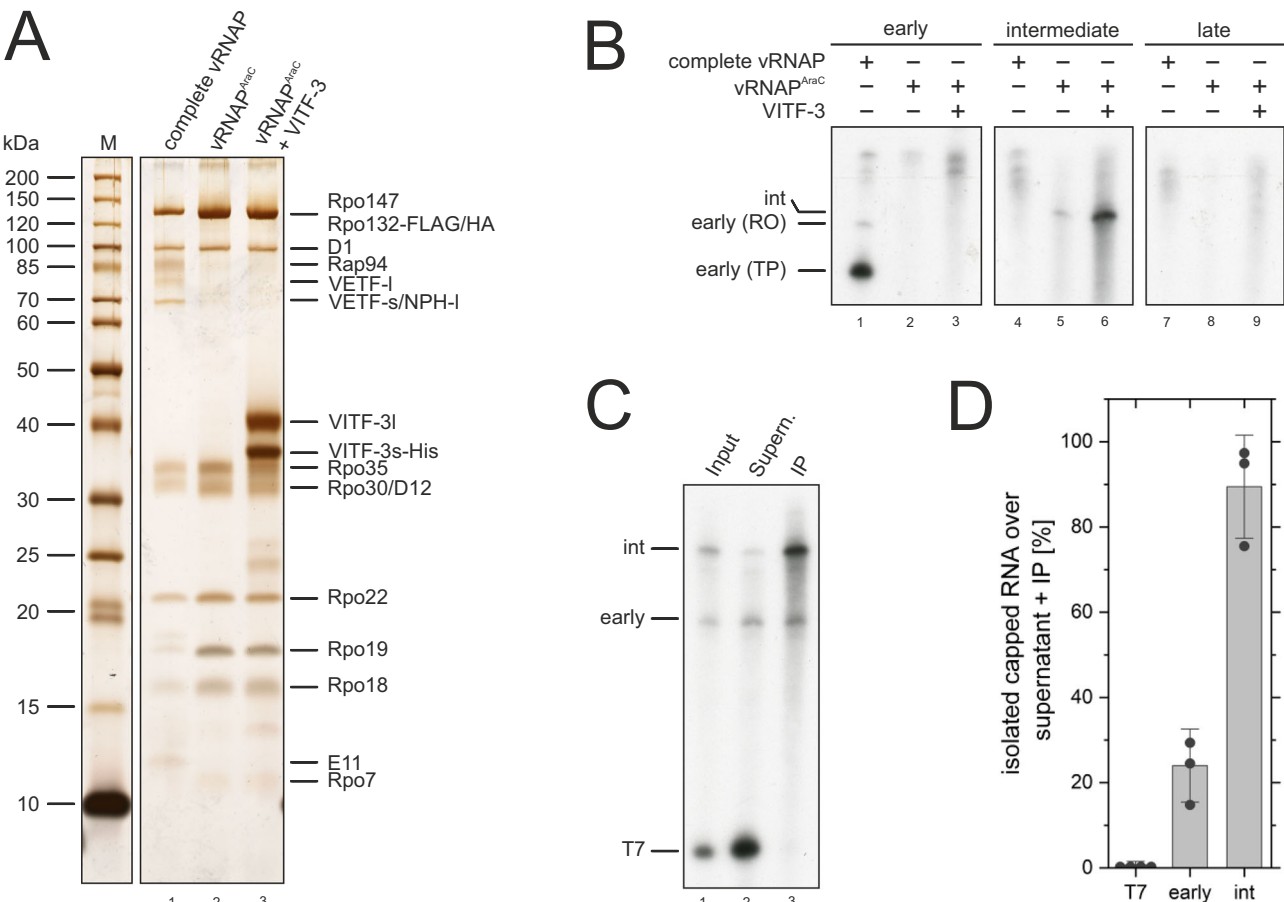

**Fig. 4 | vRNAP, CE and VITF-3 enable intermediate transcription and co-transcriptional capping in vitro. A** Protein composition of vRNAP complexes used for transcription assays. Complete vRNAP (lane 1), vRNAP[AraC] (lane 2) and vRNAP[AraC] with recombinant VITF-3 (lane 3) were separated by SDS-PAGE and visualized by silver staining. A molecular weight marker and protein names are indicated on the left and right side, respectively. Source data are provided as a Source data file. **B** Analysis of the promoter specificity provided by different vRNAP complexes. The vRNAP complexes shown in (A) were tested on early, intermediate (*G8R*) and late (*A9L*) promoter-containing templates in in vitro transcription assays[44,68]. The composition of the assay is indicated on top, and the size of RNA products is indicated on the left, whereas early transcription yields a properly terminated product (TP) and a run-off product (RO). Source data are provided as a Source data file. **C** Analysis of the 5′-end of in vitro transcribed intermediate RNAs in respect to m[7]GpppN cap modifications. Transcripts of early, intermediate and T7 reactions were pooled and immunoprecipitated with the m[7]G-cap specific monoclonal antibody H-20[54]. Immunoprecipitated RNAs were separated by PAGE and visualized via radioautography. Source data are provided as a Source data file. **D** Quantification of signal intensities from (**C**). Bars represent means ± SD of at least three biological replicates (dots). Source data are provided as a Source data file.

both domains[21]. As a result, the ePIC differs significantly from archaeal and eukaryotic PICs, both in the trajectory of upstream DNA and in VETF-l simultaneously contacting the DNA and vRNAP[5,20]. In Vaccinia intermediate transcription we now found VITF-3l holding a tripartite TBP fold, which manifests in an additional i-lobe and expansion segment giving VITF-3l its characteristic "U"-shape. This expanded structure enables two contact points with the TFIIB homologue VITF-3s to create a closed ring structure. Unlike TBP, VITF-3l has a large interaction surface with vRNAP at the outer side of the i-lobe domain, which is required to close the ring around the promoter. The TBP homologs bend the upstream DNA, thereby priming it for sequence readout, which is carried out by another domain of VETF-l in the ePIC or by vRNAP in the iPIC. Overall, the TBP homologs found in Vaccinia originate from a prototypic TBP fold but developed unique structural and functional features during the host–virus co-evolution. The contrasting architectures of the ePIC and iPIC illustrate how a single RNAP core can be functionally diversified through the co-evolution of stage-specific TFs. These findings not only highlight the adaptability of the viral transcription machinery but also provide a conceptual framework to understand the modular evolution of transcription systems across all domains of life.

## Discussion

In this study, we dissected intermediate-stage transcription of the prototypic poxvirus Vaccinia using a combination of biochemical enrichment, functional assays, and structural analysis. We identified the heterodimeric VITF-3 as a TBP/TFIIB pair with unique features. Unlike canonical TFs that bind the promoter prior to polymerase recruitment and PIC formation, VITF-3 does not recognize the promoter alone. Instead, it cooperates with vRNAP and CE to enable PIC formation. Furthermore, our study shows that vRNAP recognizes the intermediate promoter in a sequence-specific manner, which enables the deposition of VITF-3 onto the upstream promoter element in a clamp-loader-like fashion. It proposes a mechanism whereby vRNAP[AraC] transiently opens the VITF-3 ring and then re-closes it around the promoter DNA. This coordinated process generates the iPIC, which is primed to produce m[7]G-capped intermediate transcripts.

Previous structural investigation of poxviral transcription focused on early vRNAP complexes as they represent the predominant form of vRNAP when isolated from infected cells[20–22,56]. To investigate intermediate transcription, we established a protocol based on the inhibition of DNA replication, to selectively enrich early viral gene products, such as intermediate TFs and vRNAP subunits[51,57]. This protocol caused

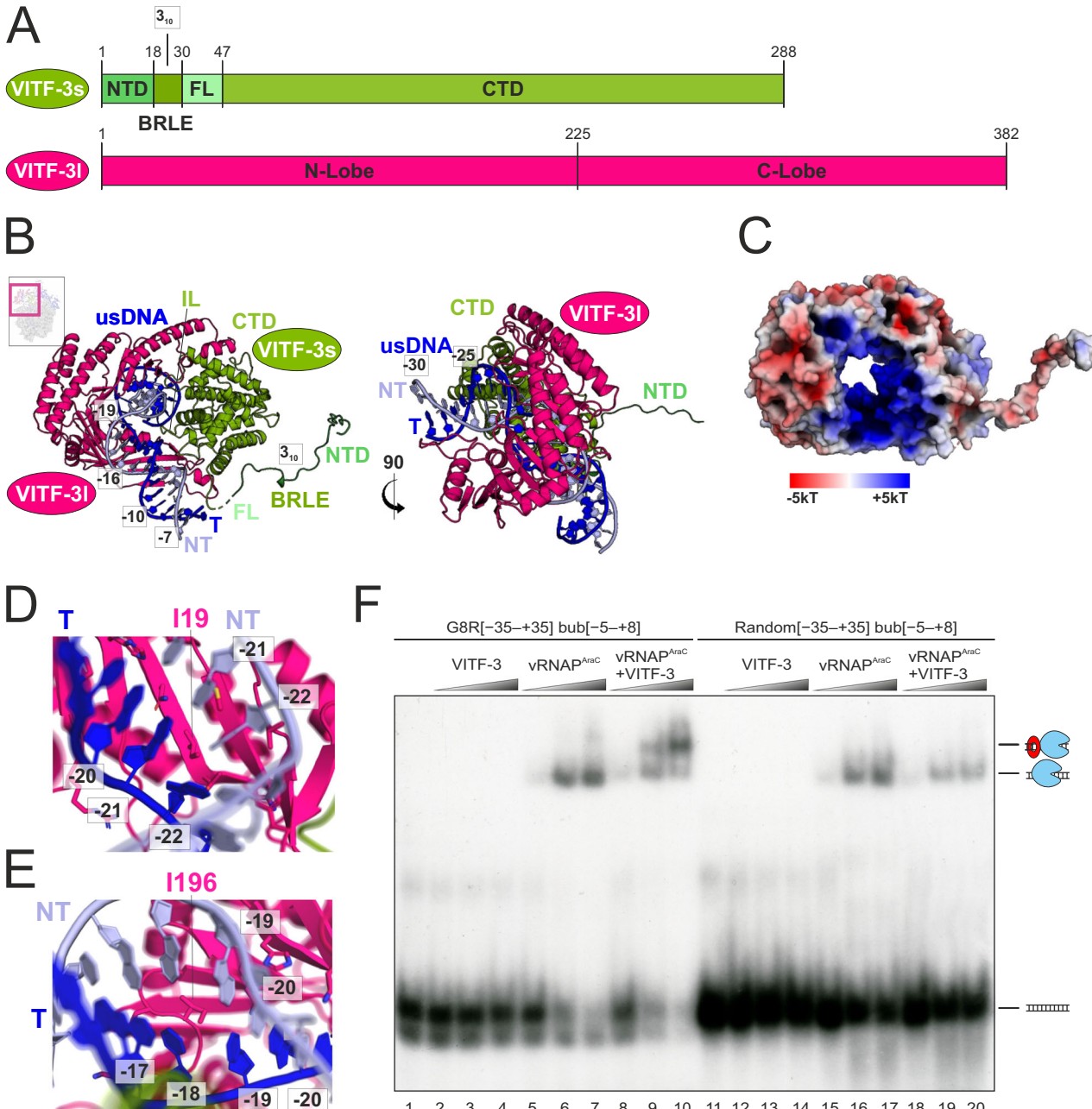

**Fig. 5 | vRNAP-assisted loading of the ring-shaped VITF-3 onto the upstream promoter. A** Domain architecture of VITF-3s and VITF-3l with functional elements indicated. NTD N-terminal domain, BRLE B-reader-like element, FL fork loop, CTD C-terminal domain. **B** Cartoon depiction of VITF-3 bound to the upstream promoter in two orthogonal views. The inserted box shows the location in the iPIC. Protein names and functional elements are shown. usDNA upstream DNA, T template strand, NT non-template strand, IL insertion loop. **C** Electrostatic Poisson−Boltzmann potential mapped on the surface of VITF-3. Blue indicates positive, and red a negative electrostatic potential. **D** Isoleucin 19 of VITF-3l intercalating into the minor groove of the promoter DNA at position −21/−22. **E** Isoleucin 196 of VITF-3l intercalating into the minor groove of the promoter DNA at position −18/−19. **F** Electrophoretic mobility shift assay using radio-labeled intermediate promoter-containing DNA and randomized DNA both holding a DNA mismatch bubble from position −5 to +8. VITF-3, vRNAP^AraC or both were added as indicated above. Schematic illustrations of formed complexes are given on the right. Source data are provided as a Source data file.

a nearly quantitative elimination of the early TFs from the purified vRNAP. However, the CE was stably bound to vRNAP under these conditions. In addition, we observed sub-stoichiometric amounts of VITF-3 as part of this complex indicating its intrinsic affinity to vRNAP or CE. Notably, the interaction of VITF-3 and vRNAP was significantly enhanced upon the addition of an intermediate promoter scaffold, which resulted in the formation of a stable iPIC.

Previous biochemical studies have provided a detailed understanding of the factors involved in intermediate transcription within cellular extracts. This process was shown to depend on VITF-3, CE, as

well as the host factors G3BP1 and p137[26,46–49]. However, our in vitro transcription assays demonstrated robust activity on intermediate templates using only a minimal set of components comprising VITF-3, CE and vRNAP. Considering that G3BP1 and p137 have previously been described to mediate the formation of phase-separated compartments such as viral factories, we propose that these factors may increase the local concentration−and potentially also the stability−of proteins involved in intermediate transcription. Thus, G3BP1 and p137 might exert an indirect stimulatory effect through their biophysical properties, rather than by acting as TFs. In contrast to the data presented

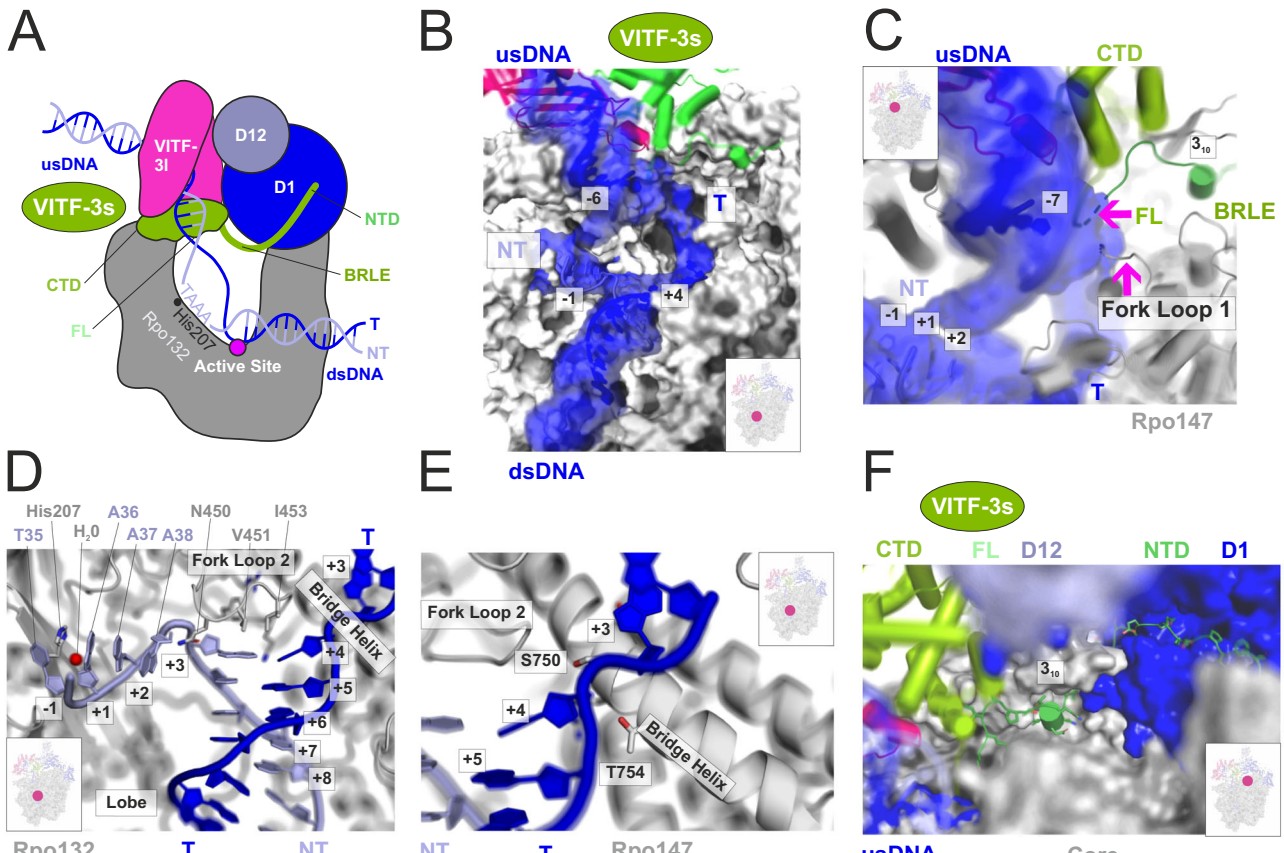

**Fig. 6 | vRNAP detects the TSS and stabilizes melted DNA strands in the iPIC.** The inserted boxes indicate the location in the iPIC structure. **A** Schematic illustration of the iPIC, wherein important features are indicated. usDNA upstream DNA, dsDNA downstream DNA, T template strand, NT non-template strand, CTD C-terminal domain, NTD N-terminal domain, FL fork loop, BRLE B-reader-like element. **B** Overview of the DNA scaffold embedded in the vRNAP cleft. The clamp has been removed for clarity and the DNA is depicted as cartoon model surrounded by its cryo-EM density. **C** DNA strand separation by the VITF-3s FL and FL 1 of Rpo147 at the trailing fork point. The DNA is depicted as cartoon model surrounded by its cryo-EM density. **D** Base-specific read-out of the TAAA motif at the leading fork point mediated by the Rpo132 lobe domain. Important domains, residues and bases are indicated. **E** Close-up view into the Rpo147 bridge helix contacting the template strand. **F** Intercalation of the VITF-3s NTD into the RNA exit channel along the outside of vRNAP and CE.

here, partially purified VITF-3 was previously shown to bind the intermediate promoter independently of vRNAP, thereby forming a committed complex[45]. However, our data suggests that VITF-3 forms a closed stable ring even in the absence of DNA, which makes unassisted binding to the promoter rather unlikely. Nevertheless, VITF-3 may remain stably bound to the promoter when vRNAP transitions into the elongation phase, thus enabling efficient re-initiation for the subsequent transcription events.

Our structural analysis of the iPIC is consistent with decades-old biochemical studies of the Vaccinia intermediate promoter, highlighting an essential AT-rich, but not sequence-conserved, core element around position −20[44,45]. This DNA motif is recognized and bent by VITF-3. As we did not observe any sequence-specific interactions between VITF-3 and its bound DNA within the iPIC, we propose that VITF-3 engages the core element through shape-based readout, sensing the physical properties of the AT-rich promoter rather than specific nucleotide identities. The spacer element spanning base pairs −14 to −2 is not involved in direct protein contacts, but its length of approximately one DNA helical turn is important for exact spatial alignment of VITF-3 and vRNAP$^{AraC}$ on the promoter elements to form the iPIC. At the TSS, the TAAA motif of the initiator element is crucial for transcription initiation, since these four bases are directly recognized by vRNAP. Notably, a similar sequence-specific recognition mechanism has been previously described for RNAP III transcription termination highlighting the co-evolution of viral and eukaryotic

transcription systems[58]. We note, however, that the TAAA motif appears with high frequency in the Vaccinia genome even outside of intermediate promoters. VITF-3 may hence contribute additional promoter-selectivity by recognizing shape and flexibility of the core element.

The availability of high-resolution functional ePIC and iPIC structures enabled their comparison to other known PICs formed by multi-subunit RNAPs and provides insights into the evolution of transcription initiation. Despite extensive divergence, each Vaccinia PIC contains a set of proteins with structurally conserved TBP-like and TFIIB-like folds[20]. Due to minimal sequence conservation, these factors were uncovered only by structural analysis or advanced bioinformatic approaches. Their conserved functionality—the TBPLF binding the upstream DNA, bending and melting the promoter, and the TFBLP bridging between TBPLF and RNAP—mirrors a common PIC architecture observed across all archaeal and eukaryotic systems[1,5,59]. However, poxviral TBPLFs are markedly divergent owing to increased asymmetry and structural insertions, which support the ring-shaped architecture of VITF-3 in the iPIC, and further sequence adaptations that provide new interaction surfaces. This specialized architecture allows VITF-3 to encircle the core promoter element but necessitates a dedicated mechanism for TF loading. Given the size and organization of the Vaccinia genome, it appears unlikely that the ring slides over a genome end and migrates to cognate promoters, particularly since VITF-3 lacks ATPase activity. Instead, we propose transient opening of

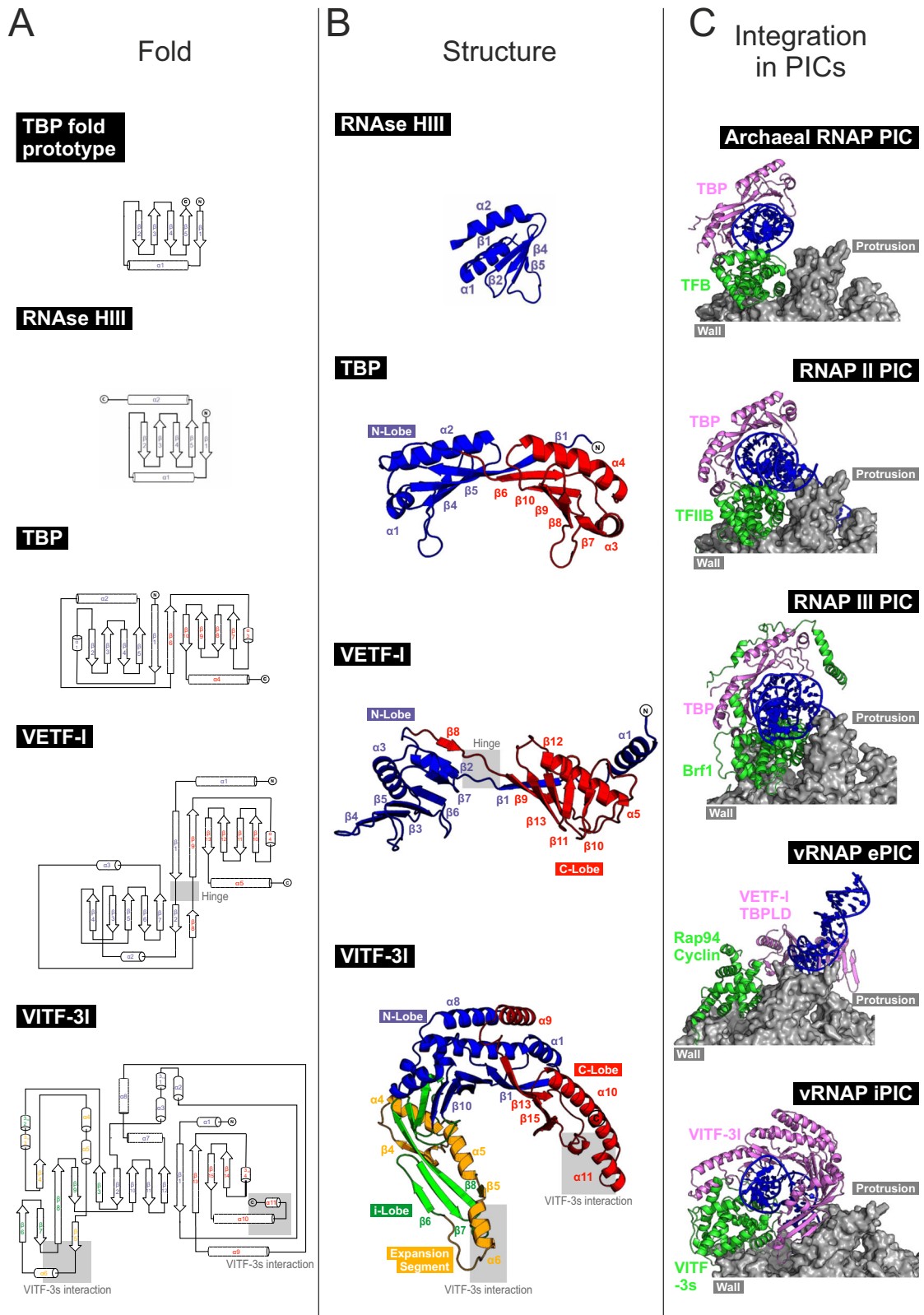

**Fig. 7 | Vaccinia transcription factors evolved from canonical TBP.**
**A** Development of the TBP secondary structure from a monopartite TBP prototype over RNAse HIII to bipartite canonical TBP and its homologs in Vaccinia. **B** Cartoon illustrations showing the structures of RNAse HIII including amio acids 1–69 (3VN5)[55], TBP (1YTB)[7], VETF-l from complete vRNAP (6RFL)[21] and VITF-3l from the iPIC$_d$ (8P0J). **C** Structural comparison of the TBP/TFIIB pairs and their homologs from PICs of archaeal RNAP (manually modeled from PDB entry 1AIS)[75], RNAP II (7EGC)[76], RNAP III (6EU0)[77], as well as the Vaccinia ePIC (7AMV)[20] and iPIC$_d$ (8P0J).

the ring that allows its loading onto the DNA, which is supported by our in vitro reconstitution experiments of the iPIC. Complementary time-resolved or single-molecule experiments could reveal mechanistic insights in VITF-3 opening and closure. Furthermore, our experiments suggest that ring opening (and subsequent closing around the DNA) is facilitated by vRNAP[AraC]. In line, our iPIC shows two slightly different conformations that likely represent snapshots of a dynamic trajectory that might inject force into the VITF-3 ring. At the same time, the two CE modules exhibit high mobility relative to each other, as observed across different transcription complexes[20]. We hence propose a dynamic scenario where the CE acts as the effector that initiates ring opening in the presence of DNA and mediates a clamp-loading reaction in cooperation with vRNAP.

Based on our study, we deduce a model for the poxviral intermediate transcription cycle. In the first step, the promoter is recognized through a concerted process involving clamp-loading of VITF-3 by vRNAP[AraC] and positioning of vRNAP at the initiator element. Because no helicases involved in promoter opening were detected, we propose that DNA melting is driven by protein–DNA interactions, particularly the clamping of VITF-3 around the promoter. These interactions bend the DNA and promote strand separation in the vRNAP cleft, leading to the formation of the iPIC. Several intermediate promoter elements are necessary for proper positioning of the transcription machinery and ensure sequence specificity. The iPIC structure is remarkably similar to an ITC or CCC with the +1 base of the template strand positioned next to the active site, which primes the complex for initial transcription[5,22]. Our study further suggests that once the nascent RNA is extended to approximately 12 nucleotides, it clashes with the N-terminus of VITF-3s obstructing the RNA exit channel, which likely results in the displacement of VITF-3 followed by promoter escape. Once loaded on the core element, VITF-3 may stay there to allow more efficient transcription re-initiation. Shortly after promoter escape, the nascent RNA reaches the CE, where it acquires a m⁷GpppN cap structure. The elongating intermediate vRNAP complex resembles the early elongation complex, although the CE may remain associated with vRNAP throughout transcription[20]. The molecular mechanism of Vaccinia intermediate transcription termination remains unknown, but an intrinsic termination mechanism appears likely since no dedicated factors have been described so far.

Our study provides insight into key mechanisms of Vaccinia intermediate transcription at atomic resolution and offers structural perspectives on the evolutionary innovations of viral transcription systems. Given the high sequence conservation of TFs among *Orthopoxviruses*, our findings are also relevant to the investigation of the human pathogens MPXV and variola virus.

## Methods

### Cell lines and growth conditions
Male African green monkey kidney fibroblasts (CV-1) were purchased from the American Type Culture Collection (ATCC No. CCL-70). Human HeLa S3 cells (female) came from Sigma-Aldrich (87110901). Both cell lines were cultivated routinely in 15 cm plates at 37 °C with 5% CO₂ in DMEM (Gibco, 41965062) supplied with 10% FBS (Gibco, 10270106) and 1% Pen/Strep (Gibco, 15140122) if not mentioned otherwise. The generation of the recombinant *Vaccinia* virus GLV-1h439 strain was described previously[21].

### Infection of HeLa S3 cells with recombinant Vaccinia virus GLV-1h439
For the infection in absence or presence of AraC (Sigma-Aldrich, C1768), the medium of 80% confluent HeLa S3 cells cultivated in 15 cm dishes was changed to DMEM supplied with 10% FBS, 1% Pen/Strep and 250 µg/ml AraC[51,52,57,60]. After 1 h of incubation, the medium was changed to DMEM supplied with 2% FBS, 1% Pen/Strep and 250 µg/ml AraC, before cells were infected with genetically modified Vaccinia virus

GLV-1h439 (Genelux Corporation), expressing FLAG/HA-tagged Rpo132, at an MOI of 10. After 1 h of incubation, DMEM supplied with 18% FBS, 1% Pen/Strep and 250 µg/ml AraC was added to generate a final FBS concentration of 10% and the cells were further incubated at standard conditions. 24 h after infection, the cells were detached from the plate using a cell scraper and centrifuged at 1800 × g and 4 °C for 10 min. After discarding the supernatant, the cell pellet was flash-frozen in liquid nitrogen and stored at −80 °C. To purify complete vRNAP, HeLa S3 cells were infected at an MOI of 2 over 48 h using the same virus in absence of AraC[21,50].

### Purification of FLAG-tagged vRNAP complexes
Infected HeLa S3 cells were thawed on ice and resuspended in lysis buffer (50 mM HEPES, pH 7.5, 150 mM NaCl, 1.5 mM MgCl₂, 0.5% (v/v) NP-40, 1 mM DTT, 10.9 µM leupeptin, 1.5 µM aprotinin, 0.2 mM PMSF and 0.1 mM AEBSF). After centrifugation at 48,000 × g for 40 min, the supernatant was incubated for 3 h at 4 °C with 200 µl anti-FLAG M2 affinity gel (Sigma-Aldrich, A2220). The beads were washed four times with buffer containing 50 mM HEPES, pH 7.5, 150 mM NaCl, 1.5 mM MgCl₂, 0.1% (v/v) NP-40 and 1 mM DTT and once with elution buffer (50 mM HEPES, pH 7.5, 150 mM NaCl, 1.5 mM MgCl₂ and 1 mM DTT). The bound protein was eluted from beads with elution buffer supplemented with 200 µg/ml 3 x FLAG peptide (Sigma-Aldrich, F4799) followed by concentrating using a Vivaspin 500 (Sartorius, VS0142) according to the vendor's protocol. For the purification of vRNAP[AraC], aliquots were flash-frozen in liquid nitrogen and stored at −80 °C. To sufficiently purify complete vRNAP, the eluate was further purified via 10% to 30% sucrose density gradient centrifugation at 194,000 × g and 4 °C for 16 h[21]. 200 µl fractions were collected, analyzed via SDS-PAGE and fractions containing complete vRNAP were pooled. After concentrating with a Vivaspin 500 (Sartorius, VS0142) according to the vendor's protocol, aliquots were prepared, flash-frozen in liquid nitrogen and stored at −80 °C.

### Purification of the Vaccinia iPIC
HeLa S3 cells infected with GLV-1h439 in the presence of AraC were thawed on ice, before resuspending in lysis buffer (150 mM NaCl, 50 mM HEPES pH 7.5, 1.5 mM MgCl₂, 0.5% (v/v) NP-40, 1 mM DTT, 10.9 µM leupeptin, 1.5 µM aprotinin, 0.2 mM PMSF and 0.1 M AEBSF) and incubation at 4 °C for 20 min. The lysate was cleared by centrifugation at 48,000 × g and 4 °C for 40 min. The supernatant was supplied with ATP (Boehringer Mannheim, 519979) to a final concentration of 4 mM as well as with 10 nmol of annealed *G8R* promoter DNA containing an artificial mismatch bubble from position −5 to +8 (Template strand: 5′CCCCCTTTATGGATTTTTATAGGGATGGAGTAAA ATATAAATTTGTAAATTATTTAAAGTTAAATGGCTGC3′, Non-template strand: 5′GCAGCCATTTAACTTTAAATAATTTACAAAAATTTAAAATGA GCATCCCTATAAAAATCCATAAAGGGGG3′). The sample was incubated at 20 °C for 30 min under rotation, before the vRNAP complex was purified via FLAG-IP as described in the section above. The elution fractions were pooled and concentrated in a Vivaspin 500 (Sartorius, VP0142) followed by 5% to 45% sucrose density gradient centrifugation at 194,000 × g and 4 °C for 16 h[21]. 200 µl fractions were collected before they were analyzed via SDS-PAGE and silver staining. The fractions containing co-migrating vRNAP[AraC] and endogenous VITF-3 (fractions 11–14) were pooled and concentrated in a Vivaspin 500 (Sartorius, VP0142) by centrifugation at 8000 × g and 4 °C to a concentration of 1.5 µg/µl. During concentration, additional elution buffer was applied to remove residual sucrose. Afterwards, the sample was centrifuged at 15,000 × g and 4 °C for 5 min, before the supernatant was used for grid preparation. Quantifoil R1.2/1.3 Au 300 grids were glow-discharged for 90 s at medium intensity using the Plasma Cleaner model PDC-002 (Harrick Plasma Ithaca, NY/USA). Per grid, 3 µl sample were applied at 4 °C and 100% humidity using a Vitrobot Mark IV instrument (FEI Company). The settings used included 0 s wait time, 5 s

blotting time and a blot force of 20. The grids were stored in liquid nitrogen till data collection.

## Cryo-EM structure determination and model building

The Cryo-EM dataset of the iPIC sample was collected with a Thermo Fisher Titan Krios G3 equipped with a Falcon III camera (Thermo-Fischer). Data was acquired with EPU at 300 keV and a primary magnification of 75,000 (calibrated pixel size 1.0635 Å) in movie-mode with 25 fractions per movie and integrating the electron-signal. The total exposure was 70 e⁻/Å over an exposure time of 4.5 s with 2 exposures per hole. The dataset was processed with Cryosparc[61]. Two cycles of 2D classification and manual selection of classes based on the appearance of their class averages for clean-up resulted in a stack of 1,421,158 good particles. An ab-initio map was created from a 50,000 particles subset and the full set subjected to a consensus 3D refinement. The set was then separated into three classes that represented the iPIC in three slightly different conformations (Table 1, Supplementary Fig. 2A). The three particle sets were finally subjected to a non-uniform refinement step, and the density was docked with the previously published core vRNAP model (6RIC [https://doi.org/10.2210/pdb6RIC/pdb]) and the coordinates of the two CE modules extracted from the complete vRNAP model (6RFL [https://doi.org/10.2210/pdb6RFL/pdb])[21]. The remaining density was identified as VITF-3s and VITF-3l and docked with an AlphaFold 2 prediction of both polypeptides. The resulting model was then further manually adjusted in COOT (version 0.9.8.95) and automatically refined with Phenix.real_space_refine including an ADP refinement step[62,63]. Secondary structure, and mild Ramachandran restraints were imposed. A total of three cycles of manual inspection and automated refinement were performed.

## Single-pot, solid-phase-enhanced sample preparation for MS

Samples were processed using an adapted SP3 protocol[64]. In brief, the reconstitution solution was added to a final volume of 125 µl. DTT was added to a concentration of 5 mM, and the alkylation was performed with 20 mM iodoacetamide. 10 mM additional DTT were used for quenching. Equal volumes of two types of Sera-Mag Speed Beads (Cytiva, 45152101010250 and 65152105050250) were combined, washed with water and 5 µL of the bead mix were added to each sample. 140 µl 100% ethanol were added and the samples were incubated at 24 °C for 5 min under shaking at 1000 rpm. The beads were immobilized on a magnetic rack for 2 min, and the supernatant was removed, before two subsequent wash steps with 200 µl 80% ethanol (Chromasolv, Sigma) and one wash step with 1 ml 80% ethanol. The digest was performed on beads with 0.25 µg Trypsin (Gold, Mass Spectrometry Grade, Promega) and 0.25 µg Lys-C (Wako) in a total volume of 100 µl holding 100 mM ammonium bicarbonate at 37 °C overnight. Peptides were desalted using C-18 Stage Tips[65]. Each Stage Tip was prepared with three discs of C-18 Empore SPE Discs (3 M) in a 200 µl pipette tip. Peptides were eluted with 60% acetonitrile in 0.1% formic acid, dried in a vacuum concentrator (Eppendorf), and stored at −20 °C. Peptides were dissolved in 2% acetonitrile/0.1% formic acid prior to nanoLC-MS/MS analysis.

## NanoLC-MS/MS analysis

NanoLC-MS/MS analysis was performed on an Orbitrap Fusion (Thermo Scientific) equipped with a PicoView Ion Source (New Objective) and coupled to an EASY-nLC 1,000 (Thermo Scientific). The peptides were loaded on a trapping column (2 cm × 150 µm ID, PepSep) and separated on a capillary column (30 cm × 150 µm ID, PepSep), both packed with 1.9 µm C18 ReproSil and separated by a linear gradient from 3% to 30% acetonitrile and 0.1% formic acid over 120 min with a flow rate of 500 nl/min. Both MS and MS/MS scans were acquired in an Orbitrap analyzer with a resolution of 60,000 for MS scans and 30,000 for MS/MS scans. HCD fragmentation with 35% normalized collision

energy was applied. A Top Speed data-dependent MS/MS method with a fixed cycle time of 3 s was used. Dynamic exclusion was applied with a repeat count of 1 and an exclusion duration of 90 s, whereas singly charged precursors were excluded from selection. The minimum signal threshold for precursor selection was set to 50,000. Predictive AGC was used with a target value of $4 \times 10^5$ for MS scans and $5 \times 10^4$ for MS/MS scans. EASY-IC was used for internal calibration.

## MS data analysis

Raw MS data files were analyzed with MaxQuant version 1.6.2.2[66]. The database search was performed with Andromeda, which is integrated in the utilized version of MaxQuant. We searched against the UniProt Human Proteome database (Release 2022_02, UP000005640, 79,864 proteins) and the UniProt Vaccinia Virus Proteom (Release 2021_09, 257 proteins). Additionally, a database containing common contaminants was used. The search was performed with tryptic cleavage specificity which allowed for 3 mis-cleavages. Protein identification was under control of the false-discovery rate (FDR; <1% FDR on protein and peptide spectrum match (PSM) level). In addition to MaxQuant default settings, the search was performed against following variable modifications: Protein N-terminal acetylation, Gln to pyro-Glu formation (N-term. Gln) and oxidation (Met). Carbamidomethyl (Cys) was set as fixed modification. Further data analysis was performed using R scripts developed in-house. LFQ intensities were used for protein quantitation[67]. Proteins with less than two razor/unique peptides were removed. Missing LFQ intensities were imputed with values close to the baseline. Data imputation was performed with values from a standard normal distribution with a mean of the 5% quantile of the combined log10-transformed LFQ intensity and a standard deviation of 0.1. The vertical axis was normalized on the Rpo132-FLAG/HA amount since this vRNAP subunit was the target for FLAG-IPs after infections with and without AraC. Three biological replicates were performed, and only viral proteins detected in at least two replicates were shown.

## Expression and purification of recombinant VITF-3 in *E. coli*

The ORFs of the Vaccinia *A8R* and *A23R* genes encoding VITF-3s and VITF-3l, respectively, were PCR-amplified from viral genomic DNA using respective primer pairs (*A8R*-5′-GGGCGGGATCCATGTTCGAAC-CAGTACCAGATC-3′, *A8R*-5′-CCGGCAAGCTTCTAAGTAAAATATTTTAG TAGCGTATCC-3′, *A23R*-5′-GAACTCATATGATGGATAATCTATTTACC TTTCTAC-3′, *A23R*-5′-GAACTGCGGCCGCTCATTTTAGAAGCAATTCT TTTAG-3′). The PCR product containing the *A8R* ORF as well as the vector pET28a were digested with BamHI and HindIII (Thermo Scientific, ER0051 and ER0502). Equally, the *A23R* PCR product and the vector pET21a were digested with NdeI and NotI (Thermo Scientific, ER0585 and ER0591). The digested DNA was purified via agarose gel electrophoresis followed by DNA recovery using a NucleoSpin Gel and PCR Clean-up kit (Macherey-Nagel, 740609) according to the vendor's protocol. The digested ORFs of *A8R* and *A23R* were ligated into the digested vectors pET28a and pET21a using T4-DNA ligase (NEB, M0202S) to generate *A8R*-pET28a and *A23R*-pET21a. Chemically competent BL21(DE3)pLysS cells (Promega Corporation, L1195) were co-transformed with 15 ng of both plasmids in a 50 µl reaction, whereas kanamycin, ampicillin and chloramphenicol resistances were used for selection. About 100 ml SB medium supplied with respective antibiotics were inoculated with BL21(DE3)pLysS cells co-transformed with *A8R*-pET28a and *A23R*-pET21a and incubated at 37 °C and under shaking at 140 rpm overnight. The next day, 2 l of SB medium distributed in two 5 l baffled flasks were inoculated with 1% of overnight pre-culture each and the cells were incubated at 37 °C under shaking at 100 rpm. After reaching an OD600 of 0.5, protein expression was induced by adding IPTG to a final concentration of 0.5 mM. The bacteria were further incubated overnight at 16 °C under shaking. After collecting cells by centrifugation at 2800 × *g* at 4 °C for 30 min, about

40 ml buffer L (25 mM HEPES pH 8.0, 150 mM NaCl, 15% glycerol, 5 mM β-mercaptoethanol) supplied with 10 mM imidazole, 10.9 μM leupeptin, 1.5 μM aprotinin, 0.2 mM PMSF and 0.1 M AEBSF were added. The cells were resuspended and lysed by sonication using a 250 Power Supply (Branson Ultrasonics, Danbury, USA) with the following settings: duty cycle 50%, output control 70%, and 5× 60 s with 60 s breaks in between. The lysate was cleared upon centrifugation at $48,000 \times g$ and 4 °C for 120 min and the supernatant was incubated with 4 ml of equilibrated His60 Ni superflow resin (Takara, 635660) overnight at 4 °C under rotation. The beads were subsequently washed with 20 ml buffer L supplied with 20 mM imidazole, 40 ml buffer L supplied with 20 mM imidazole and 1 M NaCl, 20 ml buffer L supplied with 20 mM imidazole and 40 ml buffer L supplied with 50 mM imidazole. For elution, the beads were incubated with 4 ml buffer L supplied with 500 mM imidazole for about 30 min at 4 °C under rotation. The elution fraction was collected, and the procedure was repeated two times, before the eluates were combined and filtered using a 0.2 μm ReliaPrep Syringe Filters (Ahlstrom-Munksjö Germany GmbH, 760516). The Ni-NTA eluate was further purified via a 1 ml HiTrap Heparin HP affinity column (Cytiva, 17040601) equilibrated with buffer A (25 mM HEPES pH 8.0, 150 mM NaCl, 15% glycerol, 1 mM DTT). After applying the Ni-NTA eluate at a flow rate of 0.2 ml/min, impurities were removed by a first elution step using buffer A supplemented with 350 mM NaCl. The protein of interest was eluted in buffer A with 1 M NaCl in a second elution step. 0.5 ml fractions were collected and analyzed via SDS-PAGE, before fractions containing pure VITF-3 were pooled and concentrated with a Vivaspin 500 (Sartorius, VP0142) to 20–30 mg/ml. Aliquots were flash-frozen in liquid nitrogen and stored at −80 °C.

## In vitro transcription assay

In vitro transcription was carried out in 25 μl reactions containing 40 mM Tris pH 8.0, 6 mM MgCl₂, 10 mM DTT, 2 mM spermidine, 80 μM S-adenosyl methionine (NEB, B9003S), 20 U RNase Inhibitor (NEB, M0314S, M0314L), 2.5 mM ATP/GTP/CTP (Thermo Scientific, R0481), 0.5 mM UTP and 33.3 nM $[\alpha\text{-}^{32}P]$-UTP (Hartman Analytic, SCP-410). The template for Vaccinia early transcription, was pSB24, which was described previously[68]. For intermediate and late template generation, the *G8R* promoter ranging from −178 to +501 and the *A9L* promoter from −139 to +417 were PCR amplified using respective primer pairs (*G8R*-5'-GAACTGAATTCCCGATTCCTTTTTCGATGC-3', *G8R*-rev-5'-GAACTGGATCCCTGATGCTGACTAATACACTTG-3', *A9L*-fw-5-GAACTGAATTCGTCAAGTCTTACTCATTGATCCG-3', *A9L*-rev-5'-GAACTGGATCCGAATTATTTACAAAATTCATTAACGAG-3'). The PCR products as well as the pUC19 vector were digested with BamHI and EcoRI (Thermo Scientific, ER0051 and ER0271), before the promoters were ligated into pUC19 by T4 DNA Ligase (NEB, M0569S) to generate *G8R*-pUC19 and *A9L*-pUC19. The plasmids pSB24, *G8R*-pUC19 and *A9L*-pUC19 were proliferated in XL1blue cells, before linearization at positions +631, +750 and +666, respectively, using NdeI (Thermo Scientific, ER0585). Per transcription reaction, 500 ng of linearized plasmid, 0.5 μg vRNAP$^{AraC}$ and 0.5 μg recombinant VITF-3 were combined, before the sample was incubated at 30 °C for 45 min. To stop the reaction, 25 μl water, 200 μl TRIzol reagent (Sigma-Aldrich, 15596026) and 50 μl chloroform were added. After mixing, the sample was centrifuged at $12,000 \times g$ at 4 °C for 15 min, before the aqueous phase was transferred into a fresh tube containing 200 μl 2-propanol and 1 μl GlycoBlue Coprecipitant (Invitrogen, AM9516). The solutions were mixed, and the RNA was precipitated at −20 °C for at least 10 h. The sample was centrifuged at $12,000 \times g$ and 4 °C for 15 min and the supernatant was discarded, before the RNA pellet was washed with 180 μl of 70% ethanol. The pellet was dried at 37 °C for 5 min and resuspended in 20 μl 1x RNA loading dye (47.5% formamide, 0.025% bromophenol blue, 0.025% xylene cyanole). After heating up to 95 °C for 5 min, 10 μl of sample were loaded on a 4% polyacrylamide gel containing 8 M urea. Electrophoresis was carried out at 20 mA in 1x

TBE running buffer. After 120 min, the electrophoresis was stopped, the gel was transferred onto Whatman paper and exposed to an Amersham Hyperfilm MP (Cytiva, 28-9068-44) at −80 °C. The film was developed using an OPTIMAX X-Ray Film Processor (PROTEC GmbH & Co KG, Oberstenfeld, Germany).

## Analysis of m⁷G-capping of in vitro transcribed RNA

For the investigation of the 5'-end of in vitro transcribed intermediate RNA, early and intermediate RNAs were generated in 125 μl transcription reactions each. Additionally, a third reaction containing 50 μl of T7 transcribed U1 snRNA served as non-capped control[69]. After precipitation with isopropanol, all RNAs were resuspended in 30 μl water and pooled. 100 μl of Dynabeads Protein G (Invitrogen, 10004D) were washed twice with PBS containing 0.02% Tween20. The beads were incubated with 60 μg H-20 antibody (Reinhard Lührmann) in 400 μl PBS supplied with 0.02% Tween20 for 45 min under shaking at room temperature[54]. After removing unbound antibody, the combined transcripts were added to the beads after taking off 20% for input. The sample was incubated for 70 min under rotation at RT, before the supernatant, which contained unbound RNAs, was taken off. The beads were washed three times with 400 μl PBS supplied with 0.02% Tween20 each, before 50 μl water were added. The RNAs of input, unbound and IP samples were extracted via TRIzol reagent (Sigma-Aldrich, 15596026). Therefore, all samples were filled up with water to 50 μl, before four times the volume TRIzol was added. After mixing, 50 μl chloroform were added and the samples were centrifuged at $12,000 \times g$ for 15 min at RT. The aqueous phase was transferred into a fresh tube containing four times the volume 2-propanol and 1 μl GlycoBlue Coprecipitant (Invitrogen, AM9516). After mixing, the RNA was precipitated at −20 °C for at least 10 h. The sample was centrifuged at $12,000 \times g$ at 4 °C for 15 min and the supernatant was discarded, before the RNA pellet was washed once with 180 μl 70% ethanol. The RNA pellet was dried at 37 °C for 5 min and resuspended in 20 μl 1x RNA loading dye (47.5% formamide, 0.025% bromophenol blue, 0.025% xylene cyanole). After boiling the sample at 95 °C for 5 min, 3 μl of input and supernatant, as well as 4 μl of bead sample were loaded on a 4% polyacrylamide gel containing 8 M urea. The sample volumes varied between biological replicates. Electrophoresis was carried out at 20 mA in 1x TBE running buffer. After 120 min, electrophoresis was stopped, and the gel was transferred onto Whatman paper and exposed to an Amersham Hyperfilm MP (Cytiva, 15596026) at −80 °C. The film was developed after 7 h using an OPTIMAX X-Ray Film Processor (PROTEC GmbH & Co KG, Oberstenfeld, Germany), before signal intensities were quantified using ImageJ[70]. The signal intensity of the IP fraction was divided by the sum of IP and supernatant, representing the ratio of capped RNA to total RNA. OriginPro 2023 (v10.0.0.154, OriginLab Corporation, Northampton, MA, USA) was used to visualize bars representing mean values ± SD of at least three independent replicates (dots).

## 5'-labeling of DNA

DNA oligonucleotides containing the *G8R* promoter or a randomized DNA sequence from −35 to +35 with or without a DNA mismatch bubble from −5 to +8 positions in respect to the TSS (Sigma-Aldrich), were dissolved in 1x Annealing buffer (20 mM HEPES pH 7.5, 100 mM NaCl, 3 mM MgCl₂) to a concentration of 100 μM. A list containing all DNA scaffolds used is provided in Supplementary Table 1. To add a radioactive 5'-phosphate on the single-stranded DNA oligonucleotide, a T4-Polynukleotid-Kinase (Thermo Scientific, EK0031) reaction containing 0.17 μM $[\alpha\text{-}^{32}P]$-ATP (Hartmann Analytics, SRP-501) was incubated at 37 °C for 60 min. The sample was filled up to 20 μl with water, before the labeled DNA was purified via a MicroSpin G-25 Column (Cytiva, 27532501) according to the vendor's protocol. A small portion of the purified sample was taken off to serve as ssDNA sample, whereas the remaining template strand DNA was combined with equal amounts of respective non-labeled, non-template strand. For annealing, the

sample was incubated at 95 °C for 15 min, before the sample continuously cooled down to RT. After 5 h, the annealed DNA was diluted to 500 fmol/µl in water and used for EMSAs.

### Electrophoretic mobility shift assay

EMSAs were carried out in 10 µl reactions containing 10 mM Tris pH 8.5, 5 mM MgCl$_2$, 1 mM DTT, 1 mM ATP/GTP/UTP/CTP (Thermo Scientific, R0481), and 0.5, 1,0 or 1.5 pmol VITF-3 and vRNAP$^{AraC}$. Last, 500 fmol of 5′-labeled DNA were added and the sample was incubated at 30 °C for 30 min. If indicated, antibodies targeting the HA tag (BioLegend, 901516) or VITF-3 (immunoGlobe Antikörpertechnik GmbH, Himmelstadt, Germany) were added and the sample was further incubated at 30 °C for 15 min. 5x native dye (50% glycerol, 2,5x TBE, 0.05% bromophenol blue) was added to the sample, before it was applied on a native 4% polyacrylamide gel with an acrylamide to bisacrylamide ratio of 37.5:1. Electrophoresis was carried out at 120 V in 0.25x TBE running buffer. After 4 h, electrophoresis was stopped, the gel was transferred onto Whatman paper and exposed to an Amersham Hyperfilm MP (Cytiva, 15596026) at −80 °C. The film was developed using an OPTIMAX X-Ray Film Processor (PROTEC GmbH & Co KG, Oberstenfeld, Germany).

### Generation of a sequence logo of the Vaccinia intermediate promoter

The intermediate promoter logo was generated using the MEME program (version 5.5.8)[71]. To do so, 48 intermediate promoter sequences ranging from −50 to +7 in respect to the TSS were extracted from the Vaccinia genome (NCBI Reference Sequence: NC_006998.1) and used for consensus calculations[52,53]. One motif was generated with a length between 35 and 38 bp. The approach was adopted from Yang et al. (2011) and Deng et al. (2025)[52,53].

### Sequence analysis and AlphaFold predictions of VITF-3 subunits

To identify homologs of both VITF-3 subunits, the Vaccinia protein sequences (UniProt Q80HV2 and P68720) were used individually as queries in Phmmer to search the UniProt database for related proteins[72]. Relevant hits included *Orthopoxvirus* members−Vaccinia virus (UniProt A0A6B7KEZ4 and A9J0Q7), Cowpox virus (UniProt Q80DV2 and Q8QMU2), Taterapox virus (UniProt Q0NP66 and Q0NP82), Variola virus (UniProt Q0N518 and Q89179), and Monkeypox virus (UniProt Q3I8M9 and Q3I7J8)−as well as the Nile crocodile poxvirus (UniProt Q070A5 and Q070C2). The identified VITF-3 sequences were imported into Jalview (version 2.11.5.0, The Barton Group, University of Dundee, Scotland, UK), and multiple sequence alignments were generated separately for each subunit[73]. Structural predictions for the Vaccinia and Nile crocodile virus VITF-3 complexes were subsequently obtained using AlphaFold 3[74].

### Quantification and statistical analysis

For the quantification of the capping efficiencies of different RNAPs, the signal intensities obtained in supernatant and IP fractions were quantified using ImageJ[70]. The signal intensity of IP was divided by the sum of IP and supernatant, representing the ratio of capped RNA to total RNA in %. OriginPro 2023 (v10.0.0.154, OriginLab Corporation, Northampton, MA, USA) was used to visualize the data. The bars represent mean values ± SD of at least three independent replicates (dots).

### Reporting summary

Further information on research design is available in the Nature Portfolio Reporting Summary linked to this article.

## Data availability

The coordinate files generated in this study have been deposited in the Protein Data Bank under accession codes 8POJ (iPIC$_d$), 8PON (iPIC$_s$), and 8POK (iPIC$_{CEm}$). The cryo-EM density data generated in this study have been deposited in the Electron Microscopy Data Bank under accession codes EMD-17334 (iPIC$_d$), EMD-17336 (iPIC$_s$), and EMD-17335 (iPIC$_{CEm}$). The structural data used in this study are available in the Protein Data Bank under accession codes 7AMV (ePIC)[20], 6RIE (CCC)[22], 3VN5 (RNase HIII)[55], 1YTB (TBP/TATA-box complex)[7], 6RFL (complete vRNAP)[21], 6RIC (core vRNAP)[21], 1AIS (TBP/TFB core/TATA-box complex from *Pyrococcus woesei*)[75], 7EGC (RNAP II PIC)[76], and 6EU0 (RNAP III PIC)[77]. The mass spectrometry proteomics data have been deposited to the ProteomeXchange Consortium via the PRIDE partner repository with the dataset identifier PXD065561. Source data are provided with this paper.

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

## Acknowledgements
We would like to thank Reinhard Lührmann for kindly providing the H-20 antibody. Cryo-EM of the iPIC was carried out in the cryo-EM facility of the Julius-Maximilians University Würzburg (Projects INST 93/903-1 #359471283, INST 93/1042-1 #456578072, INST 93/1143-1 x525040890). This work was supported by a DFG grant to UF and CG (Fi 573/27-1) and by Volkswagen-Stiftung (9B813).

## Author contributions
U.F., C.G., J.B., and S.J. conceived the study. S.J. performed complex reconstitution and purification as well as biochemical assays. S.L. and A.S. conducted and analyzed the mass spectrometry experiments. C.G. and S.J. prepared cryo-EM samples and collected the data. C.G. carried out data processing, model building, structural analysis, and visualization. U.F., C.G., and S.J. wrote the manuscript. U.F. and C.G. acquired funding.

## Funding

## Competing interests
The authors declare no competing interests.
