## [Transparent Peer Review file · Nature Communications]

Cooperative clamp-mediated promoter recognition by poxviral RNA polymerase and its TBP/TFIIB-like partner

Corresponding Author: Professor Utz Fischer

Version 0:

Reviewer comments:

Reviewer #1

(Remarks to the Author)

Overall Comments

This manuscript presents a detailed structural and biochemical analysis of the intermediate pre-initiation complex (iPIC) in poxvirus transcription, focusing on the cooperative role of the viral RNA polymerase (vRNAP) and the TBP/TFIIB-like factor VITF-3. The study provides significant insights into the unique mechanism of promoter recognition and transcription initiation in poxviruses, highlighting the evolutionary divergence and specialization of transcription machinery in these viruses. The combination of cryo-EM structural analysis, *in vitro* transcription assays, and biochemical experiments strengthens the conclusions drawn by the authors. The findings not only advance our understanding of poxvirus transcription but also offer a comparative perspective on transcription initiation mechanisms across different systems.

Main Points and Suggestions

1. The claim that vRNAP acts as a clamp loader for VITF-3 is central to the study. While the structural data offer a static snapshot of VITF-3 in a ring-shaped conformation around the promoter, the dynamic process of how vRNAP facilitates the opening and closing of this ring requires more robust experimental support. Complementary time-resolved or single-molecule experiments could offer real-time insights into the loading mechanism and strengthen this aspect of the study.
2. It's also important to enhance the analysis of how VITF-3 distinguishes between different AT-rich sequences. Though the preference of VITF-3 for AT-rich regions is noted, the sequence recognition mechanism isn't fully explored. A more comprehensive investigation into the binding affinities of VITF-3 with various AT-rich sequences, possibly through mutagenesis studies and detailed sequence analysis, would provide a clearer picture of its sequence-specificity.
3. Regarding the evolutionary analysis of VITF-3, the current comparison with canonical TBP is a good starting point, but could be expanded. Incorporating a broader range of viral sequences into the analysis would offer a more complete view of the evolutionary trajectory and functional divergence of VITF-3. This would involve examining sequence conservation and variation across different poxviruses and related viruses, shedding light on key residues and structural motifs important for their unique function.

Minor Points

1. While the methods section is comprehensive, some experimental details (e.g., exact buffer conditions for cryo-EM grid preparation) could be expanded slightly for reproducibility.
2. The manuscript occasionally uses specialized terminology (e.g., "clamp-loader-like mechanism") that may not be familiar to all readers. Simplifying or defining such terms upon first use would enhance accessibility.
3. Minor grammatical inconsistencies (e.g., hyphenation in compound terms) should be addressed during revision.

Reviewer #2

(Remarks to the Author)
Review of Jungwirth et al., 2025

General points. This excellent article by the Fischer team presents the structural basis of Vaccinia virus intermediate transcription initiation complexes (iPICs) solved by cryoEM. NCLDV viruses encode a pol II-like enzyme but unlike in cellular transcription, the viral RNAP engages with distinct sets of initiation factors to facilitate the temporal control of transcription, during early, intermediate and late infection stages of the virus. This structural and functional plasticity is unique and only found in NCLDV RNAPs; rationalising how the same RNAP can utilise three different sets of initiation factors is a complex challenge that can only be addressed by meticulously solving the molecular structures of early, intermediate and late pre-initiation complexes (PICs). The factors are not homologous and provide idiosyncratic solutions to the mechanical engineering aspects of transcription initiation. Surprisingly, structural features of the basal factors TBP and TFIIB became apparent within the very large VETFI/s and Rap94 factors in the context of early Vaccinia preinitiation complexes (ePICs) solved by the Fischer group.

This article by Fischer and colleagues is focussed on intermediate complexes and reports that the two subunits of the heterodimeric intermediate factors VITF-3 are highly divergent TBP/TFIIB factors. They form a ring structure around the promoter DNA, which has never been observed with TBP/TFIIB factors before – fascinating stuff! Moreover, the molecular mechanisms of promoter recognition are highly unusual as the RNAP functions as clamp loader of the VITF-3 ring, and the RNAP itself contributes to sequence-specific interactions.

Personally, I've been waiting for this discovery for years, and it is eagerly anticipated in the community, the intermediate complex is a blind spot no more! The structural data are of very high quality, the body text tackles the description and comparison of different PIC structures both elegantly and eloquently, the mechanistic working hypotheses of intermediate gene transcription initiation based on the structures are rigorous and not overinterpreted; I recommend publication in your journal without further delay.

Specific points.

I.35/38 I would make it clear that all RNAPI, II & III utilise TBP and TFIIB-related factors (based on functional assays), this sentence gives the appearance that RNAPI is not as its not resolved in cryoEM structures (I.44)

I.40 TBP-TATA interactions do not lead to DNA strand separation/melting (requires TFIIB)

I.51 And the TFIIB linker becomes part of the 'composite' active site by stabilising the template DNA strand in the active site prior to RNA synthesis

I.81 what is the role of the host factors G3BP1 and p137? To validate the statement made in I.344 (i.e. that as iPIC has robust activity, the host factors only play a supportive role), it would be important to include the two factors in the experiments. Do they stably associate with the iPIC? Do they change its topology? Do they boost the transcription activity 100 fold? We'll never know until its tested. Publication of the paper does not depend on these experiments, if its not feasible, the authors could extend their considerations about possible functions, expression patterns etc in the discussion section of the paper.

I.90 A 'biochemical dissection' in its true meaning would require a perturbation the factors (by mutagenesis), identifying the role of individual factor domains by deleting them or by SDM followed by a functional analysis, assembly and in vitro transcription. I think the correct term here is 'structural dissection' as its based on the molecular structure.

I.241 Replace 'by the core transcriptional components' with 'by the Vaccinia RNAP'

I.249 I don't understand the rationale for the 'concerted loading mechanism'; how does vRNAP-AraC 'cooperatively open the ring to load it onto' the promoter? What is the mechanism of DNA melting in iPICs? Apparently no helicase/translocase like VETFI or NPH-I required for DNA strand separation or loading of the template strand into the active site, so I assume the energy contributions required for DNA melting are provided by binding events (bit like the archaeal PIC)?

L.253 onwards: it would be helpful to include a figure with a sequence alignment of Vaccinia intermediate promoter sequences to emphasise the conservation of the DNA motifs/signatures.

I.287 onwards: concerning the early evolution of TBP folds, ref 45 describes RNase A. I'm not sure that RNase A is the most likely ancestor of TBP, rather it is RNase HIII which fulfils that role. May I recommend another paper with a stricter focus on TBP fold evolution that is more the point in this context (Nucleic Acids Research, 2013, 1–14, doi:10.1093/nar/gkt045, Evolutionary history of the TBP-domain superfamily).

I like the excellent discussion section of the paper which takes the wider perspective nicely into account. Maybe the authors could share their thoughts on the following? What is the fate of VITF-3 after the RNAP has escaped the immediate promoter? Does it remain associated with the promoter, or does it 'tag' along with the RNAP? Do the two VITF-3 subunits dissociate after release, or once-loaded remain a heterodimer in the cell?

Again, congrats on a beautiful piece of structural biology, I'm looking forward to its publication!

Reviewer #3

(Remarks to the Author)

Cooperative clamp-mediated promoter recognition by poxviral RNAP and its TBP/TFIIB-like partner

Jungwirth et al (Utz Fischer)

This is an excellent study that sheds light on a distinctive mechanism of transcription initiation by vaccinia RNAP at an intermediate viral promoter. The authors cleverly enrich for an intermediate transcription complex containing intermediate

factor VITF3 by affinity purification of RNAP from araC-treated infected cells and then assemble a PIC on an intermediate promoter DNA with a melted bubble at the initiation site. They proceed to solve cryo-EM structures of the complex.

The key insight is that the VITF3 heterodimer forms a ring-shaped clamp around the DNA duplex upstream of the start site and strongly kinks the DNA. Similarities and differences versus early transcription complexes reported previously by the Fischer lab are succinctly highlighted.

Additional results establish that VITF3 per se does not bind the intermediate bubble-containing promoter but joins after binding of RNAP.

There is a thoughtful discussion of the evolution of TBP, VETFL, and VITF3L folds and an interesting hypothesis that capping enzyme associated with RNAP aids loading of the VITF3 clamp.

Requested revisions/suggestions:

1) The Abstract is redundant with respect to repeating the claims of “previously unknown mode of promoter recognition”(line 22) and the “thus far unknown transcription initiation” (line 29). I suggest the authors simply close the Abstract with “. . . propose vRNAP as a clamp loader for VITF-3.” [In light of the discussion in which they speculate that CE associated with RNAP aids in VITF3 clamp loading, do they want to include this point in the closing sentence of the Abstract?]

2) In citing Baroudy & Moss 1980 (ref 26) for purifications of vRNAP from virions (line 59), the authors neglected to cite the contemporaneous study: Spencer et al (1980) Purification and properties of vaccinia virus DNA-dependent RNA polymerase. *J. Biol. Chem.* 255, 5388-5395.

3) In referencing vaccinia transcription termination and the involvement of capping enzyme, the authors neglected to cite the discovery of this mechanism by the Moss lab: Shuman, Broyles, Moss (1987) Purification and characterization of a transcription termination factor from vaccinia virions. *J. Biol. Chem.* 262, 12372-12380.

4) In citing the requirement for Rap94 (the product of the H4 gene) in early transcription initiation (line 68), the authors can also cite: Deng et al (1994) A role for the H4 subunit of vaccinia RNA polymerase in transcription initiation at a viral early promoter. *J. Biol. Chem.* 269, 14323-14329.

5) In citing Harris, Rosales, Moss 1993 (ref 36; line 74) for the fact that capping enzyme engages elongating RNAP for cotranscriptional capping and termination, the authors neglected to cite the following studies that establish the points regarding cotranscriptional capping (Hagler et al (1992) A freeze-frame view of eukaryotic transcription during elongation and capping of nascent mRNA. *Science* 255, 983-986) and the fact that the guanylyltransferase and methyltransferase activities of capping activity are not required for termination factor activity (Luo et al (1995) The D1 and D12 subunits are both essential for the transcription termination factor activity of vaccinia virus capping enzyme. *J. Virol.* 69, 3852-3856.)

6) Line 254: “previous chapter” is awkward. “section” would be a better choice.

7) The authors state in Discussion that “we show that VITF-3 forms a closed stable ring even in the absence of DNA” (line 348) but they do not report a structure of VITF3 alone. Can they clarify the basis for the claim (or did I miss something?).

Version 1:

Reviewer comments:

Reviewer #1

(Remarks to the Author)

The authors have adequately addressed my questions and alleviated my concerns.

Reviewer #2

(Remarks to the Author)

All good!

Reviewer #1 (Remarks to the Author)

Overall Comments

This manuscript presents a detailed structural and biochemical analysis of the intermediate pre-initiation complex (iPIC) in poxvirus transcription, focusing on the cooperative role of the viral RNA polymerase (vRNAP) and the TBP/TFIIB-like factor VITF-3. The study provides significant insights into the unique mechanism of promoter recognition and transcription initiation in poxviruses, highlighting the evolutionary divergence and specialization of transcription machinery in these viruses. The combination of cryo-EM structural analysis, in vitro transcription assays, and biochemical experiments strengthens the conclusions drawn by the authors. The findings not only advance our understanding of poxvirus transcription but also offer a comparative perspective on transcription initiation mechanisms across different systems.

Main Points and Suggestions

1. The claim that vRNAP acts as a clamp loader for VITF-3 is central to the study. While the structural data offer a static snapshot of VITF-3 in a ring-shaped conformation around the promoter, the dynamic process of how vRNAP facilitates the opening and closing of this ring requires more robust experimental support. Complementary time-resolved or single-molecule experiments could offer real-time insights into the loading mechanism and strengthen this aspect of the study.

Thank you for this insightful suggestion. Indeed, we were also thinking about using biophysical methods such as FRET analysis to monitor the VITF-3 clamp-loading in a time-resolved manner. However, since we can obtain active VITF-3 only by co-expression of both subunits in *E. coli*, we currently do not have the possibility to fluorescently label its subunits individually. Moreover, when Sanz *et al.* discovered VITF-3 as Vaccinia intermediate TF in 1998/1999, they stated “Moreover, transcription activity was not detected when the two ORFs were expressed separately, and the proteins were mixed together” (Sanz *et al.*, 1999). This suggests that the subunits cannot be handled separately but rather must be co-expressed. Because of these technical limitations the suggested experiment is very challenging and not feasible in the time frame of the revision of this manuscript.

2. It's also important to enhance the analysis of how VITF-3 distinguishes between different AT-rich sequences. Though the preference of VITF-3 for AT-rich regions is noted, the sequence recognition mechanism isn't fully explored. A more comprehensive investigation into the binding affinities of VITF-3 with various AT-rich sequences, possibly through mutagenesis studies and detailed sequence analysis, would provide a clearer picture of its sequence-specificity.

Indeed, the sequence recognition preference of VITF-3 is an important aspect that has been examined previously by Baldick *et al.* (1992) and Knutson *et al.* (2006) through single-nucleotide substitutions within the intermediate promoter. To provide a broader view of promoter conservation, we generated a sequence logo of the Vaccinia intermediate promoter based on 48 intermediate genes, as described by Deng *et al.* (2025) and Yang *et al.* (2011). This analysis has been included as a new subpanel in Figure 1.

Our EMSA data of the originally submitted manuscript had demonstrated that VITF-3 does not bind the promoter alone, whereas vRNAP^{AraC} shows weak interaction with the mismatched DNA scaffold. Furthermore, vRNAP^{AraC} is required for VITF-3 loading, so we propose a concerted binding mechanism of VITF-3 and vRNAP^{AraC} to their respective promoter regions. Because both the core and initiator elements are important for protein binding, we carried out additional EMSAs using promoter templates with randomized core or

initiator element sequences. In both cases, iPIC formation was strongly impaired. Moreover, introducing either element individually into a randomized promoter did not restore complex formation. These results further underscore the importance of cooperative binding by VITF-3 and vRNAP^{AraC} to their cognate promoter elements. The corresponding experiments have been added as a new subpanel in Supplementary Figure 3.

3. Regarding the evolutionary analysis of VITF-3, the current comparison with canonical TBP is a good starting point, but could be expanded. Incorporating a broader range of viral sequences into the analysis would offer a more complete view of the evolutionary trajectory and functional divergence of VITF-3. This would involve examining sequence conservation and variation across different poxviruses and related viruses, shedding light on key residues and structural motifs important for their unique function.

This is a very interesting topic we have not looked at before. As suggested, we searched for related proteins in the UniProt database using Phammer. Homologs of VITF-3 were generally found in the *Poxviridae* sub-family of *Chordopoxvirinae* (infecting vertebrates), but no homologs were found in the closely related *Entomopoxvirinae* (infecting insects) or any other virus or organism in the database. VITF-3 thus seems to be an exclusive "invention" of the *Chordopoxvirinae* lineage (see new supplementary Figure 4).

In addition, we performed an AlphaFold structure prediction of VITF-3 from the Nile crocodile poxvirus, which is the most distant known homologue to date. Intriguingly, although the VITF-3s and VITF-3l sequences share only 32% and 41% identity between Vaccinia virus and Nile crocodile poxvirus, their predicted structures are nearly identical. These results were also included in the new supplementary Figure 4.

Minor Points

1. While the methods section is comprehensive, some experimental details (e.g., exact buffer conditions for cryo-EM grid preparation) could be expanded slightly for reproducibility.

Thank you for pointing out the lack of this important technical detail. We added this information to the method section.

2. The manuscript occasionally uses specialized terminology (e.g., "clamp-loader-like mechanism") that may not be familiar to all readers. Simplifying or defining such terms upon first use would enhance accessibility.

Thank you for this remark. We added brief explanations for several specialized terms throughout the manuscript to improve clarity.

3. Minor grammatical inconsistencies (e.g., hyphenation in compound terms) should be addressed during revision.

We corrected the hyphenation throughout the manuscript.

Reviewer #2 (Remarks to the Author)

General points. This excellent article by the Fischer team presents the structural basis of Vaccinia virus intermediate transcription initiation complexes (iPICs) solved by cryoEM. NCLDV viruses encode a pol II-like enzyme but unlike in cellular transcription, the viral RNAP engages with distinct sets of initiation factors to facilitate the temporal control of transcription, during early, intermediate and late infection stages of the virus. This structural and functional

plasticity is unique and only found in NCLDV RNAPs; rationalising how the same RNAP can utilise three different sets of initiation factors is a complex challenge that can only be addressed by meticulously solving the molecular structures of early, intermediate and late pre-initiation complexes (PICs). The factors are not homologous and provide idiosyncratic solutions to the mechanical engineering aspects of transcription initiation. Surprisingly, structural features of the basal factors TBP and TFIIB became apparent within the very large VETFI/s and Rap94 factors in the context of early Vaccinia preinitiation complexes (ePICs) solved by the Fischer group.

This article by Fischer and colleagues is focussed on intermediate complexes and reports that the two subunits of the heterodimeric intermediate factors VITF-3 are highly divergent TBP/TFIIB factors. They form a ring structure around the promoter DNA, which has never been observed with TBP/TFIIB factors before – fascinating stuff! Moreover, the molecular mechanisms of promoter recognition are highly unusual as the RNAP functions as clamp loader of the VITF-3 ring, and the RNAP itself contributes to sequence-specific interactions. Personally, I've been waiting for this discovery for years, and it is eagerly anticipated in the community, the intermediate complex is a blind spot no more! The structural data are of very high quality, the body text tackles the description and comparison of different PIC structures both elegantly and eloquently, the mechanistic working hypotheses of intermediate gene transcription initiation based on the structures are rigorous and not overinterpreted; I recommend publication in your journal without further delay.

Specific points.

I.35/38 I would make it clear that all RNAPI, II & III utilise TBP and TFIIB-related factors (based on functional assays), this sentence gives the appearance that RNAPI is not as its not resolved in cryoEM structures (I.44)

We rewrote this section to make the role of TBP in eukaryotic transcription clearer.

I.40 TBP-TATA interactions do not lead to DNA strand separation/melting (requires TFIIB)

We corrected this mistake.

I.51 And the TFIIB linker becomes part of the 'composite' active site by stabilising the template DNA strand in the active site prior to RNA synthesis

We added this information.

I.81 what is the role of the host factors G3BP1 and p137? To validate the statement made in I.344 (i.e. that as iPIC has robust activity, the host factors only play a supportive role), it would be important to include the two factors in the experiments. Do they stably associate with the iPIC? Do they change its topology? Do they boost the transcription activity 100 fold? We'll never know until its tested. Publication of the paper does not depend on these experiments, if its not feasible, the authors could extend their considerations about possible functions, expression patterns etc in the discussion section of the paper.

This is an interesting point that we also found puzzling when performing our study. To address this topic, we first examined the effect of lysate from uninfected HeLa S3 cells, which should contain any potential host-cell factors. The lysate enhanced intermediate transcription activity by approximately 25% (Fig. 1A), a considerably milder effect than expected, particularly given that these factors have been described as essential intermediate TFs by Katsafanas & Moss (2004) (Fig. 1B).

In addition, we tested recombinantly expressed G3BP1 and p137 in our intermediate transcription reactions *in vitro* (Fig. 1C). Whereas we could confirm in this assay that the addition of cell lysate slightly stimulates transcription (Fig. 1C, compare lanes 1 and 2), the addition of G3BP1 and p137 was slightly inhibitory rather than stimulatory (Fig 1C, lanes 3–5). However, we want to emphasize that we consider these experiments preliminary as the recombinant proteins used still contained impurities and their functionality requires further validation by additional biophysical and functional assays.

Considering that G3BP1 and p137 have previously been described to mediate the formation of phase-separated compartments (see *Yang et al., 2020*), we propose that these factors may increase the local concentration—and potentially also the stability—of proteins involved in intermediate transcription within viral factories. Thus, we suggest that G3BP1 and p137 exert *in vivo* an indirect stimulatory effect through their biophysical properties, rather than by acting as TFs.

We have included our hypothesis regarding the stimulatory role of G3BP1 and p137 in intermediate transcription into the Discussion section. If desired, the results summarized in Fig. 1 can be included as an additional supplementary figure.

Figure 1: The effect of cell lysate and G3BP1/p137 on intermediate transcription *in vitro*. (A) and (C) Autoradiography films showing radiolabeled RNA transcribed in intermediate transcription assays *in vitro*. Individual components are indicated at the top. (B) Quantification of the transcription assay shown in (A). The absolute autoradiography signal intensity was blotted against the composition.

I.90 A 'biochemical dissection' in its true meaning would require a perturbation the factors (by mutagenesis), identifying the role of individual factor domains by deleting them or by SDM followed by a functional analysis, assembly and *in vitro* transcription. I think the correct term here is 'structural dissection' as its based on the molecular structure.

We changed this phrase according to the reviewer's suggestion.

I.241 Replace 'by the core transcriptional components' with 'by the Vaccinia RNAP'

We changed this phrase as suggested.

I.249 I don't understand the rationale for the 'concerted loading mechanism'; how does vRNAP-AraC 'cooperatively open the ring to load it onto' the promoter? What is the mechanism of DNA melting in iPICs? Apparently no helicase/translocase like VETFI or NPH-I required for DNA strand separation or loading of the template strand into the active site, so I

assume the energy contributions required for DNA melting are provided by binding events (bit like the archaeal PIC)?

This is an interesting point raised by the reviewer. From functional assays we know that no helicase is involved in promoter melting, but since we used mismatched scaffolds for structural analysis, we do not have structural information about DNA melting. We know that both VITF-3 and vRNAP^{AraC} are necessary for upstream DNA strand separation since two FL domains—one of Rpo132 and one of VITF-3s—are involved in this process. As suggested by the reviewer, we also believe that the energy required for DNA melting needs to be provided by binding events, probably by the binding energy of the highly positively charged VITF-3 onto the negatively charged DNA. We consider a scenario possible wherein the loading of VITF-3 onto the upstream promoter destabilizes the DNA double helix at the TSS, which subsequently leads to binding of vRNAP^{AraC}, as well as to the melting mediated by both FL domains. However, we currently do not know how vRNAP^{AraC} can distinguish an intermediate promoter from a random AT-rich sequence in the viral genome without DNA melting. These are certainly questions we would like to address in later studies.

L.253 onwards: it would be helpful to include a figure with a sequence alignment of Vaccinia intermediate promoter sequences to emphasise the conservation of the DNA motifs/signatures.

We agree that both points are of great importance. Accordingly, we have added a motif logo of the intermediate promoter as a new subpanel in Figure 1 and refer to it later in the text. In addition, we included evolutionary analyses comprising sequence alignments of the amino acid sequences of both VITF-3 subunits in Supplementary Figure 4.

I.287 onwards: concerning the early evolution of TBP folds, ref 45 describes RNase A. I'm not sure that RNase A is the most likely ancestor of TBP, rather it is RNase HIII which fulfils that role. May I recommend another paper with a stricter focus on TBP fold evolution that is more the point in this context (Nucleic Acids Research, 2013, 1–14, doi:10.1093/nar/gkt045, Evolutionary history of the TBP-domain superfamily).

Thank you for this suggestion, we changed the TBP evolution figure accordingly.

I like the excellent discussion section of the paper which takes the wider perspective nicely into account. Maybe the authors could share their thoughts on the following? What is the fate of VITF-3 after the RNAP has escaped the immediate promoter? Does it remain associated with the promoter, or does it 'tag' along with the RNAP? Do the two VITF-3 subunits dissociate after release, or once-loaded remain a heterodimer in the cell?

We believe that VITF-3 only occurs as heterodimer independent of its "free" or promoter-bound state. Since the RNA clashes with the VITF-3s NTD at a length of about 12 nt, we think this leads to promoter escape, whereas VITF-3 stays back on the core element of the intermediate promoter. We addressed this topic more thoroughly in the discussion section.

Again, congrats on a beautiful piece of structural biology, I'm looking forward to its publication!

Reviewer #3 (Remarks to the Author)

Cooperative clamp-mediated promoter recognition by poxviral RNAP and its TBP/TFIIB-like

partner

Jungwirth et al (Utz Fischer)

This is an excellent study that sheds light on a distinctive mechanism of transcription initiation by vaccinia RNAP at an intermediate viral promoter. The authors cleverly enrich for an intermediate transcription complex containing intermediate factor VITF3 by affinity purification of RNAP from araC-treated infected cells and then assemble a PIC on an intermediate promoter DNA with a melted bubble at the initiation site. They proceed to solve cryo-EM structures of the complex.

The key insight is that the VITF3 heterodimer forms a ring-shaped clamp around the DNA duplex upstream of the start site and strongly kinks the DNA. Similarities and differences versus early transcription complexes reported previously by the Fischer lab are succinctly highlighted.

Additional results establish that VITF3 per se does not bind the intermediate bubble-containing promoter but joins after binding of RNAP.

There is a thoughtful discussion of the evolution of TBP, VETFL, and VITF3L folds and an interesting hypothesis that capping enzyme associated with RNAP aids loading of the VITF3 clamp.

Requested revisions/suggestions:

1) The Abstract is redundant with respect to repeating the claims of “previously unknown mode of promoter recognition”(line 22) and the “thus far unknown transcription initiation” (line 29). I suggest the authors simply close the Abstract with “. . . propose vRNAP as a clamp loader for VITF-3.” [In light of the discussion in which they speculate that CE associated with RNAP aids in VITF3 clamp loading, do they want to include this point in the closing sentence of the Abstract?]

We closed the abstract with “. . . propose vRNAP as a clamp loader for VITF-3.”

2) In citing Baroudy & Moss 1980 (ref 26) for purifications of vRNAP from virions (line 59), the authors neglected to cite the contemporaneous study: Spencer et al (1980) Purification and properties of vaccinia virus DNA-dependent RNA polymerase. J. Biol. Chem. 255, 5388-5395.

Thank you for the suggestion, we added this reference.

3) In referencing vaccinia transcription termination and the involvement of capping enzyme, the authors neglected to cite the discovery of this mechanism by the Moss lab: Shuman, Broyles, Moss (1987) Purification and characterization of a transcription termination factor from vaccinia virions. J. Biol. Chem. 262, 12372-12380.

Thank you for the suggestion, we added this reference.

4) In citing the requirement for Rap94 (the product of the H4 gene) in early transcription initiation (line 68), the authors can also cite: Deng et al (1994) A role for the H4 subunit of vaccinia RNA polymerase in transcription initiation at a viral early promoter. J. Biol. Chem. 269, 14323-14329.

Thank you for the suggestion, we added this reference.

5) In citing Harris, Rosales, Moss 1993 (ref 36; line 74) for the fact that capping enzyme engages elongating RNAP for cotranscriptional capping and termination, the authors neglected to cite the following studies that establish the points regarding cotranscriptional capping (Hagler et al (1992) A freeze-frame view of eukaryotic transcription during elongation and capping of nascent mRNA. *Science* 255, 983-986) and the fact that the guanylyltransferase and methyltransferase activities of capping activity are not required for termination factor activity (Luo et al (1995) The D1 and D12 subunits are both essential for the transcription termination factor activity of vaccinia virus capping enzyme. *J. Virol.* 69, 3852-3856.)

Thank you for the suggestion, we added these references.

6) Line 254: “previous chapter” is awkward. “section” would be a better choice.

We changed this phrase accordingly.

7) The authors state in Discussion that “we show that VITF-3 forms a closed stable ring even in the absence of DNA” (line 348) but they do not report a structure of VITF3 alone. Can they clarify the basis for the claim (or did I miss something?).

We tried to solve the structure of VITF-3 by cryo-EM and X-ray crystallography, but technical limitations prevented structure determination so far (too small for cryo-EM and no crystals obtained yet). Because we did not demonstrate this aspect explicitly, we have toned down the wording. Nonetheless, our combined biochemical evidence—co-expression and co-purification of both proteins and their inability to bind DNA—along with the high-confidence AlphaFold prediction strongly argues for the formation of a ring-shaped heterodimer in the absence of DNA.